# Changing temporal context in human temporal lobe promotes memory of distinct episodes

Mostafa M. El-Kalliny [1], John H. Wittig Jr[1], Timothy C. Sheehan[2], Vishnu Sreekumar[1], Sara K. Inati[3] & Kareem A. Zaghloul [1]

Memories of experiences that occur around the same time are linked together by a shared temporal context, represented by shared patterns of neural activity. However, shared temporal context may be problematic for selective retrieval of specific memories. Here, we examine intracranial EEG (iEEG) in the human temporal lobe as participants perform a verbal paired associates memory task that requires the encoding of distinct word pairs in memory. We find that the rate of change in patterns of low frequency (3–12 Hz) power distributed across the temporal lobe is significantly related to memory performance. We also find that exogenous electrical stimulation affects how quickly these neural representations of temporal context change with time, which directly affects the ability to successfully form memories for distinct items. Our results indicate that the ability to retrieve distinct episodic memories is related to how quickly neural representations of temporal context change over time during encoding.

[1] Surgical Neurology Branch, National Institute of Neurological Disorders and Stroke, National Institutes of Health, Bethesda, MD 20892, USA.
[2] Neurosciences Graduate Program, University of California, San Diego, La Jolla, CA 92093, USA. [3] Office of the Clinical Director, National Institute of Neurological Disorders and Stroke, National Institutes of Health, Bethesda, MD 20892, USA. Correspondence and requests for materials should be addressed to K.A.Z. (email: kareem.zaghloul@nih.gov)

**W**hen encoding an experience in memory, individuals rely upon the ability to link that experience to the time and place in which it occurs. This is one of the defining features of episodic memory[1]. Models that have taken into account this link between an experience and its surrounding spatiotemporal context have successfully captured many behavioral aspects of episodic memory[2,3]. For example, context can trigger the recollection of a specific experience, and conversely, recalling an experience is accompanied by retrieval of its context. Such retrieved contexts can then prompt the retrieval of other memories that were experienced at a similar time or place[3,4].

Temporal context refers to the features of an experience that occur around the time it is first experienced, and is shaped by both external inputs and a continuously changing internal state[1,2,5]. Recent evidence has demonstrated the presence of signals in the brain that may serve as an internal representation of temporal context. Individual neurons explicitly identified as time cells exhibit spiking activity that is sensitive to elapsed time following the presentation of a stimulus[6,7]. More broadly, spectral power and population spiking activity, which capture aggregate activity across multiple neural populations, have both been shown to slowly change over time[8–11]. Consistent with behavioral models of episodic memory, successful retrieval involves reinstating these gradually changing neural signals[8,9]. Moreover, recovering these internal representations of temporal context facilitates the subsequent retrieval of other memories that share a similar temporal context[8,10].

Successful memory retrieval, however, also relies upon the ability to selectively retrieve memories of specific, distinct experiences. The degree to which this is possible has been linked with how well overlapping neural representations of individual memories are orthogonalized and separated in neural space during encoding[12–14]. Circuitry within the medial temporal lobe (MTL) has been implicated in this process, which is referred to as pattern separation[15–17]. However, the inputs to circuitry responsible for pattern separation likely include both the neural representation of the memory being encoded as well as a representation of its temporal context[18]. The ability to orthogonalize memories formed around the same time may therefore be facilitated when the neural representation of each memory's temporal context is more distinct than others'. Thus, whereas shared temporal context may facilitate the retrieval of multiple memories, it may be detrimental to the selective retrieval of specific, distinct memories.

Here we test the hypothesis that the ability to selectively retrieve distinct memories is related to the extent by which representations of temporal context change over time during encoding. Using intracranial EEG (iEEG), we examine patterns of neural activity across the human temporal lobe as participants perform two verbal memory tasks which call for differing degrees of cognitive separation between memories. In the paired associates task, several distinct pairs of unrelated words are memorized in succession. During retrieval, each word pair is independently cued, and thus there is no advantage to the binding of different word pairs across time[19]. In contrast, during the free recall task, words studied together at a similar time tend to be recalled successively during memory retrieval, suggesting that the binding of words across time in this case facilitates rather than hinders memory retrieval[20,21]. By contrasting data from the two tasks, we isolate phenomena that are specifically related to the role of a gradually changing temporal context in encoding and retrieving distinct episodic memories.

## Results

**Neural drift during verbal memory tasks.** We analyzed intracranial EEG (iEEG) data from 76 participants (41 male; age 36.2 ± 1.32 years; mean±SEM) with drug resistant epilepsy who underwent a surgical procedure for placement of intracranial electrodes for seizure monitoring, and then participated in either a verbal paired associates ($n = 28$) or free recall task ($n = 48$) task (Fig. 1a, b; see Methods; Supplementary Fig. 1). Participants successfully recalled 37.0 ± 4.85% words in the paired associates task, and successfully recalled 30.4±5.14% words in the free recall task.

We examined the distributed patterns of spectral power across the temporal lobe and how they changed with time as participants progressed through each list of items in each experimental session[9]. We first focused on experimental sessions in which we passively recorded iEEG data while participants performed the tasks. We hypothesized that activity during the epochs in which there is no visual stimulus provides the most direct representation of a changing temporal context. As such, we constructed feature vectors of the distributed power across the temporal lobe using the instantaneous spectral power captured during the interstimulus epochs ($-750$ to $0$ ms before item presentation; see Methods; Fig. 1c, d). Because we had no a priori assumptions about which frequencies may contribute to the representation of temporal context, we used the spectral power across five frequency bands to construct these feature vectors. We calculated the pairwise similarity between each feature vector and averaged the similarity values for all pairs which were separated by the same amount of time, or lag. The resulting data reflect how similar the distributed pattern of power is between epochs separated by different lags.

During encoding in both the paired associates and free recall tasks, we found that the pattern of temporal lobe activity at any epoch changed when examined at a later epoch (Fig. 1e). The extent to which any pair of epochs had similar activity significantly decreased as more time elapsed between them (Spearman correlation between similarity and lags; paired associates, $r = 0.711 \pm 0.049$, $t(27) = 14.62$, $p = 2.39 \times 10^{-14}$; free recall, $r = 0.719 \pm 0.026$, $t(47) = 27.2$, $p = 2.01 \times 10^{-30}$, one-sample $t$-test).

We refer to this slowly changing pattern of spectral power as neural drift. To quantify how quickly these patterns change with time, we defined the rate of neural drift as the degree to which neural activity separated by only one lag was more similar than activity separated by two lags (Fig. 1e). A larger value for drift rate therefore reflects a more rapidly changing pattern of activity in the temporal lobe. We found that the distributions of drift rates were not significantly different from one another in paired associates and free recall ($t(74) = 0.159$, $p = 0.874$, two-sample $t$-test).

**Relations between neural drift rate and memory performance.** We were interested in understanding how the rate of drift relates to paired associates memory performance, and thus the formation of distinct memories. To establish whether neural drift at any specific frequency or time relates to memory, we calculated neural drift separately using each individual frequency and each time point, from the beginning of the interstimulus epoch to the end of the encoding epoch (see Methods). For each time-frequency combination, we correlated neural drift rate with memory performance, measured as the percentage of correctly recalled items, across lists in each experimental session (Fig. 2a). We found that the rate of drift in interstimulus 3–12 Hz activity was correlated with increased memory performance both in individual participants (example in Fig. 2a) and across all participants ($p = 0.006$,

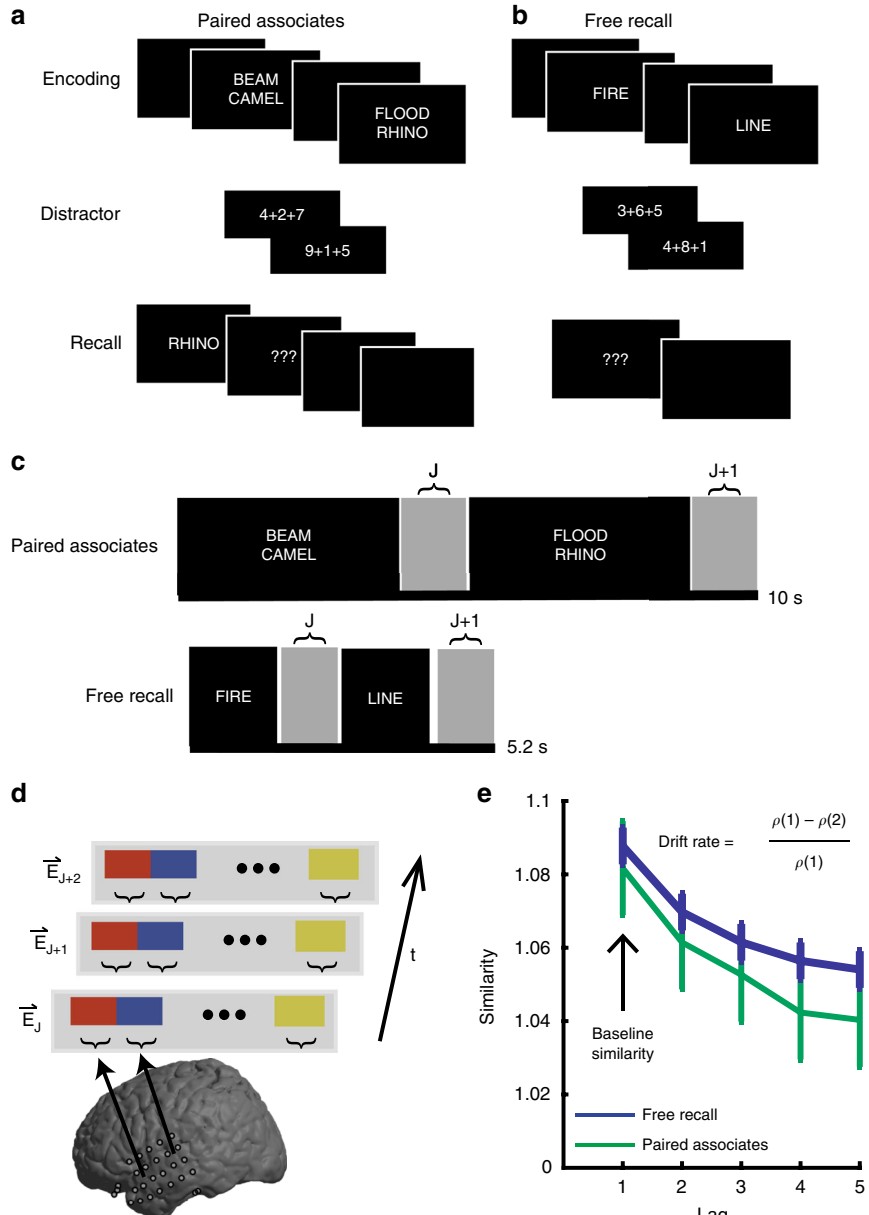

**Fig. 1** Neural drift during two verbal memory tasks. **a** During the paired associates task, pairs of words are sequentially presented on the screen. During retrieval, one word from each pair is presented in random order, and participants are instructed to vocalize the associated word. Each session consisted of up to 25 lists of this encoding-distractor-recall procedure. **b** During the free recall task, single words are sequentially presented. During retrieval, participants are instructed to vocalize remembered words, in any order. Each session consisted of up to 25 lists. **c** Interstimulus epochs are used to examine rate of neural drift during encoding. **d** For every interstimulus epoch, z-scored power in 5 frequency bands from each electrode in the temporal lobe is combined to create a single feature vector for that epoch. **e** Cosine similarity between vectors spaced apart by different lags is computed. Error bars represent Loftus-Masson confidence intervals, across 28 participants who performed the paired associates task, and 48 participants who performed the free recall task

one-sample $t$-test Fig. 2b, circled region; permutation test, see Methods). In post hoc analysis, we found a significant correlation between neural drift computed using 3–12 Hz activity in the interstimulus period ($-750$ to $0$ ms) across participants ($r = 0.095 \pm 033$, $t(27) = 2.86$, $p = 0.008$, one-sample $t$-test; Fig. 2c). We found no such relation between the overall magnitude of spectral power averaged across all temporal lobe electrodes and memory performance (Supplementary Fig. 2), suggesting that the ability to remember individual word pairs in a list improves specifically when the distributed interstimulus pattern of 3–12 Hz activity across the temporal lobe changes more quickly over time.

For our analysis, we defined the rate of neural drift as the rate by which a spatial pattern of activity changes over time. However, the baseline level of neural similarity throughout a list, which reflects the extent to which temporal context is similar throughout a list, may also be related to memory performance. We therefore repeated our analysis by examining the absolute value of cosine similarity between pairs of feature vectors describing interstimulus, 3–12 Hz activity. For each list, we defined the baseline similarity as the average value of similarity of all epochs separated by one lag (Fig. 1e). In each list, a greater baseline similarity would imply that there is greater overall

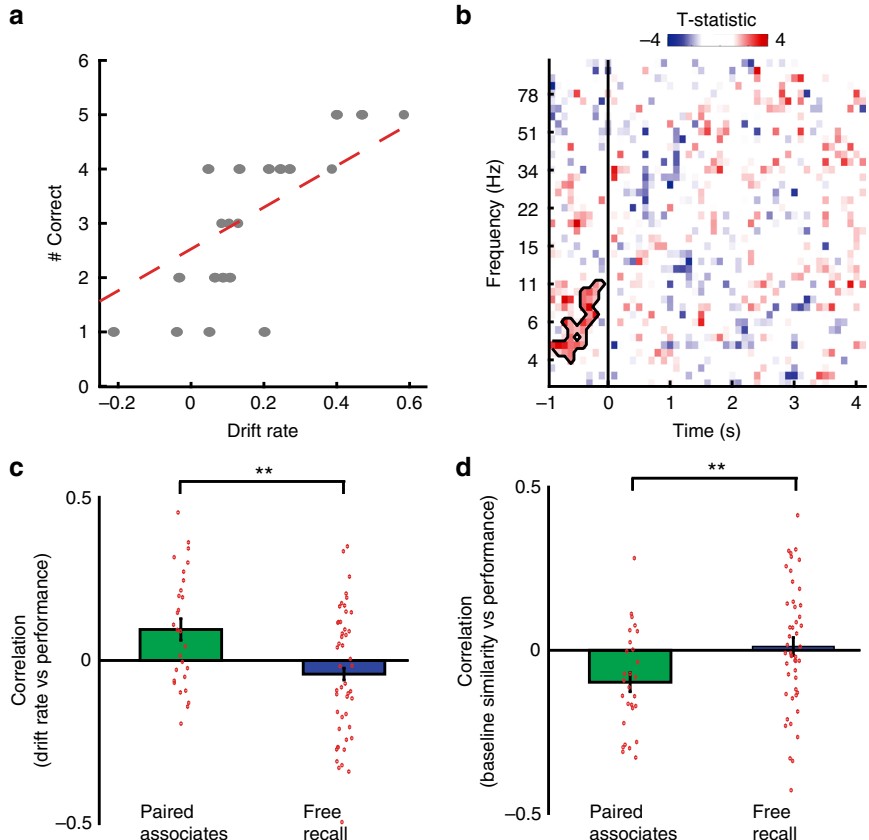

**Fig. 2** Neural drift rate relates to memory performance. **a** Correlation between performance and rate of neural drift of interstimulus 3–12 Hz activity, across all lists of a single paired associates session in an exemplar participant. **b** Correlation between paired associates performance and drift rate is computed independently for each frequency and each time epoch during encoding. Across participants, the distribution of correlation coefficients is tested using a non-parametric clustering procedure. Mask identifies the time-frequency cluster exhibiting a significant relation between memory performance and drift rate ($p = 0.006$). **c** Relationship between performance and rate of drift, as computed post-hoc using interstimulus, 3–12 Hz activity. Error bars represent SEM across 28 paired associates participants, and 48 free recall participants. *$p < 0.05$; **$p < 0.01$, $t$-test. **d** Relationship between performance and baseline similarity, as computed using interstimulus, 3–12 Hz activity. Error bars represent SEM across 28 paired associates participants, and 48 free recall participants. *$p < 0.05$; **$p < 0.01$, $t$-test

overlap in a representation of temporal context from one item to the next, across the entire list. We examined the correlation coefficient between baseline similarity and memory performance across lists in each experimental session. Across participants, we found a significant and inverse relation between baseline neural similarity and memory performance in the paired associates task ($r = 0.098 \pm 0.029$, $t(27) = 3.37$, $p = 0.002$, one-sample $t$-test; Fig. 2d), suggesting that the less similar a representation of temporal context is throughout a list, the better memory performance becomes.

The overlap in temporal context observed when memory performance is worse suggests that such overlap could lead to errors during retrieval. In the paired associates task, a direct measure of such errors is the extent to which participants retrieve the incorrect word, thereby making an intrusion. We hypothesized that greater overlap in temporal context throughout the interstimulus period, as reflected by a greater baseline similarity, would be related to greater rates of intrusions. Indeed, we found that across participants, the number of intrusions vocalized during lists with baseline similarity above the median was significantly greater than the number of intrusions vocalized during lists with baseline similarity below the median (low baseline similarity, $1.59 \pm 0.236$ intrusions, high baseline similarity, $2.18 \pm 0.202$ intrusions; $t(21) = 2.50$, $p = 0.021$, two-sample $t$-

test; $n = 22$ participants who vocalized intrusions on at least three lists).

To determine whether overlapping representations of temporal context contribute specifically to the ability to encode distinct memories, we conducted equivalent analyses on data from a separate cohort of participants who completed a free recall task. Participants that performed the paired associates task and participants that performed the free recall task exhibited similar rates of neural drift, similar electrode coverage (Supplementary Fig. 1), similar ages (paired associates, age $33.8 \pm 1.35$ years; free recall, age $36.9 \pm 1.62$ years), similar IQ levels (paired associates, WAIS IV FSIQ $89.5 \pm 4.39$; free recall, WAIS IV FSIQ $88.6 \pm 2.81$), and exhibited similar changes in low and high frequency power in the temporal lobe during successful encoding (subsequent memory effect; Supplementary Fig. 3), suggesting that both sets of participants are well matched with respect to the neural processes that underlie memory formation. However, unlike in paired associates, in free recall there is no requirement for memories to be separable from one another; instead, the binding of items across time tends to benefit memory retrieval[20,21]. We focused on neural drift rate measured using low frequency (3–12 Hz) activity during the entire interstimulus epochs ($-750$ to $0$ ms) and found that the correlation with memory performance was significantly different in the two tasks

(paired associates vs free recall, $t(74) = 3.06$, $p = 0.003$, two-sample $t$-test; Fig. 2c), although not significant in free recall alone ($t(47) = 1.49$, p = 0.143, one-sample $t$-test). Similarly, we examined the relation between the degree of baseline similarity and memory performance during the free recall task, and found that this relation was also significantly different between paired associates and free recall (paired associates vs free recall, $t(74) = 2.57$, $p = 0.012$, two-sample $t$-test; free recall, $t(47) = 0.426$, $p = 0.672$, one-sample $t$-test; Fig. 2d).

We confirmed that there was no other time-frequency cluster of activity at which the rate of neural drift was correlated with free recall memory performance (Supplementary Fig. 4). We also confirmed that the difference between paired associates and free recall was not related to the differing length of encoding epochs in the two tasks. To match the elapsed time between word pairs in the paired associates task, we repeated our analysis by defining a single unit of lag in free recall as interstimulus epochs that were separated by two intervening item presentations. We found a similar difference in the relation between drift rate and memory performance (paired associates vs free recall, $t(74) = 2.76$, $p = 0.007$, two-sample $t$-test; free recall, $t(47) = 1.47$, $p = 0.148$, one-sample $t$-test; Supplementary Fig. 5).

We hypothesized that the observed differences between paired associates and free recall would be driven by the subset of free recall participants who demonstrate a behavioral tendency to bind items across time in the service of improved memory performance. To investigate this possibility, for each participant we derived a value of temporal factor, which reflects the degree to which freely recalled items tend to be linked to one another in time, or clustered (see Methods)[21]. Participants during free recall exhibited significant temporal clustering, averaged across all recalls ($0.665 \pm 0.013$, $t(47) = 12.5$, $p = 1.37 \times 10^{-16}$, one-sample $t$-test). We divided the participants into an upper and lower tercile based on this value. The tercile of participants demonstrating the highest temporal clustering during recall ($0.759 \pm 0.019$, $n = 16$) demonstrated a relation between neural drift rate and performance that was significantly different from the relation observed during paired associates (free recall, $t(15) = 1.91$, $p = 0.076$, one-sample $t$-test; paired associates vs free recall $t(42) = 3.30$, $p = 0.002$, two-sample $t$-test; Fig. 3). Conversely, the tercile of participants demonstrating the lowest temporal clustering ($0.575 \pm 0.011$, $n = 16$) did not (free recall, $t(15) = 1.00$, $p = 0.33$, one-sample $t$-test; paired associates vs free recall, $t(42) = 1.05$, $p = 0.30$, two-sample $t$-test).

Our analysis examining the relation between drift rate and memory performance in the two tasks demonstrates that in paired associates, faster rates of neural drift in the temporal lobe specifically support the ability to retrieve separate, distinct memories. We were interested, however, in whether the rate of neural drift or the degree of baseline similarity is a global phenomenon occurring across the entire brain, or specific to individual brain regions. We therefore repeated our analyses, using the distributed pattern of 3–12 Hz power during the interstimulus period across all electrodes in the brain, in each participant. We found no significant relation between rate of neural drift across the whole brain and memory performance in either task (paired associates, $t(27) = 0.339$, $p = 0.74$, free recall, $t(47) = 1.19$, $p = 0.24$, one-sample $t$-test). Similarly, we found no significant relation between a whole-brain representation of baseline similarity and memory performance in either task (paired associates, $t(27) = 1.44$, $p = 0.161$, free recall, $t(47) = 1.48$, $p = 0.149$, one-sample $t$-test). We also examined the relation between neural drift and memory performance selectively using only patterns of activity in the frontal lobe and the parietal lobe, and found no significant relation across participants in either task (frontal lobe; paired associates, $t(21) = 0.652$, $p = 0.520$, free recall, $t(33) = 1.19$,

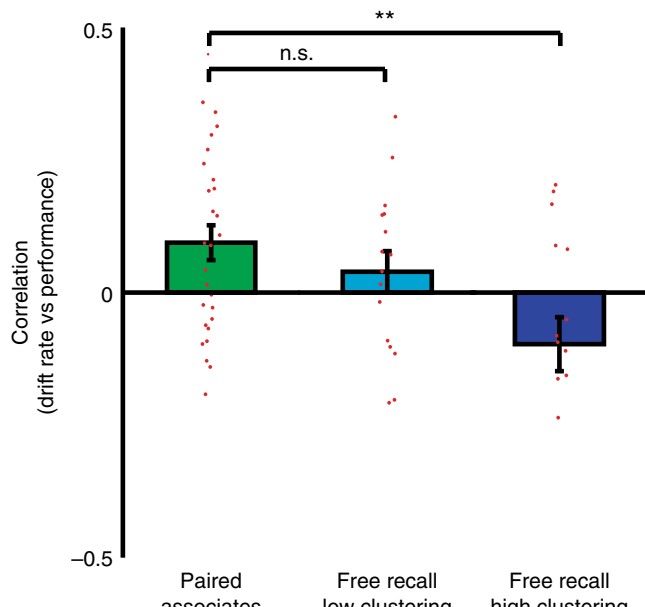

**Fig. 3** Differences between tasks, according to behavioral patterns of retrieval in free recall. For each free recall participant, we quantified the degree to which recalled items tended to be linked across time. For each participant, we computed a temporal factor by first assigning to each recall transition a number representing the degree to which each transition was temporally linked or unlinked. The temporal factor was defined as the average value for all transitions. We divided free recall participants into terciles according to their temporal factor, and compared the relation between neural drift rate and memory performance during the paired associates task to this relation in free recall independently for the upper and lower terciles. Error bars represent SEM across 28 paired associates participants, and 16 free recall participants in each group. *$p < 0.05$; **$p < 0.01$, $t$-test

$p = 0.244$; parietal lobe paired associates, $t(17) = 0.364$, $p = 0.712$, free recall, $t(28) = 0.480$, $p = 0.636$, one-sample $t$-test).

Conversely, although our data suggest that overall temporal lobe representations of temporal context underlie formation of distinct memories, it is possible that the relation between neural drift and memory performance may localize to subregions of the temporal lobe. Given previous evidence that the medial temporal lobe captures representations of a memory's spatiotemporal context[10,11], we hypothesized that the differing pattern of results in paired associates and free recall would be driven by activity of the medial temporal lobe. We therefore conducted separate analyses examining the representations of neural drift in the medial and lateral temporal lobe and their relation with memory performance. Surprisingly, we found that the relation between neural drift rate and memory performance in the paired associates task, and the difference in this relation between the two tasks, was significant for neural activity in the lateral temporal cortex (paired associates, $t(26) = 3.20$, $p = 0.0035$; free recall, $t(46) = 1.45$, $p = 0.15$, one-sample $t$-test; paired associates vs free recall, $t(73) = 3.17$, $p = 0.002$, two-sample $t$-test), but not for neural activity in the medial temporal lobe (paired associates, $t(14) = 1.08$, $p = 0.29$, free recall; $t(20) = 0.513$, $p = 0.61$, one-sample $t$-test; paired associates vs free recall, $t(35) = 0.449$, $p = 0.66$, two-sample $t$-test).

**Effects of electrical stimulation**. Our data examining passive recordings of iEEG activity reveal a relation between the rate of

neural drift and memory performance in the paired associates task. Direct electrical stimulation of the brain, however, provides an opportunity to investigate the causal nature of this relation. We therefore examined data from a subset of participants in whom we conducted closed-loop stimulation sessions in which electrical current was passed through a single pair of adjacent electrode contacts. The design of the stimulation sessions was identical to the record-only sessions, with the exception that on 11 of the 25 lists, electrical current was delivered during the encoding period (Fig. 4a, b; see Methods). For each item in a stimulation list, the decision to stimulate was controlled by a classifier trained on previous data to predict the probability of

recall on each word (see Methods). We reasoned that stimulation may disrupt the rate of neural drift during the lists in which it is applied. Electrical stimulation therefore provided an opportunity to examine whether the effect of stimulation on rate of neural drift correlates with the effect on memory performance.

For each participant receiving stimulation during the paired associates task ($N = 7$, 11 unique stimulation sites), we z-transformed the rate of neural drift in interstimulus 3–12 Hz activity and memory performance on each stimulation list, relative to the distributions of drift rate and performance observed on non-stimulation lists. Given the results observed during passive recordings of iEEG activity, we hypothesized that

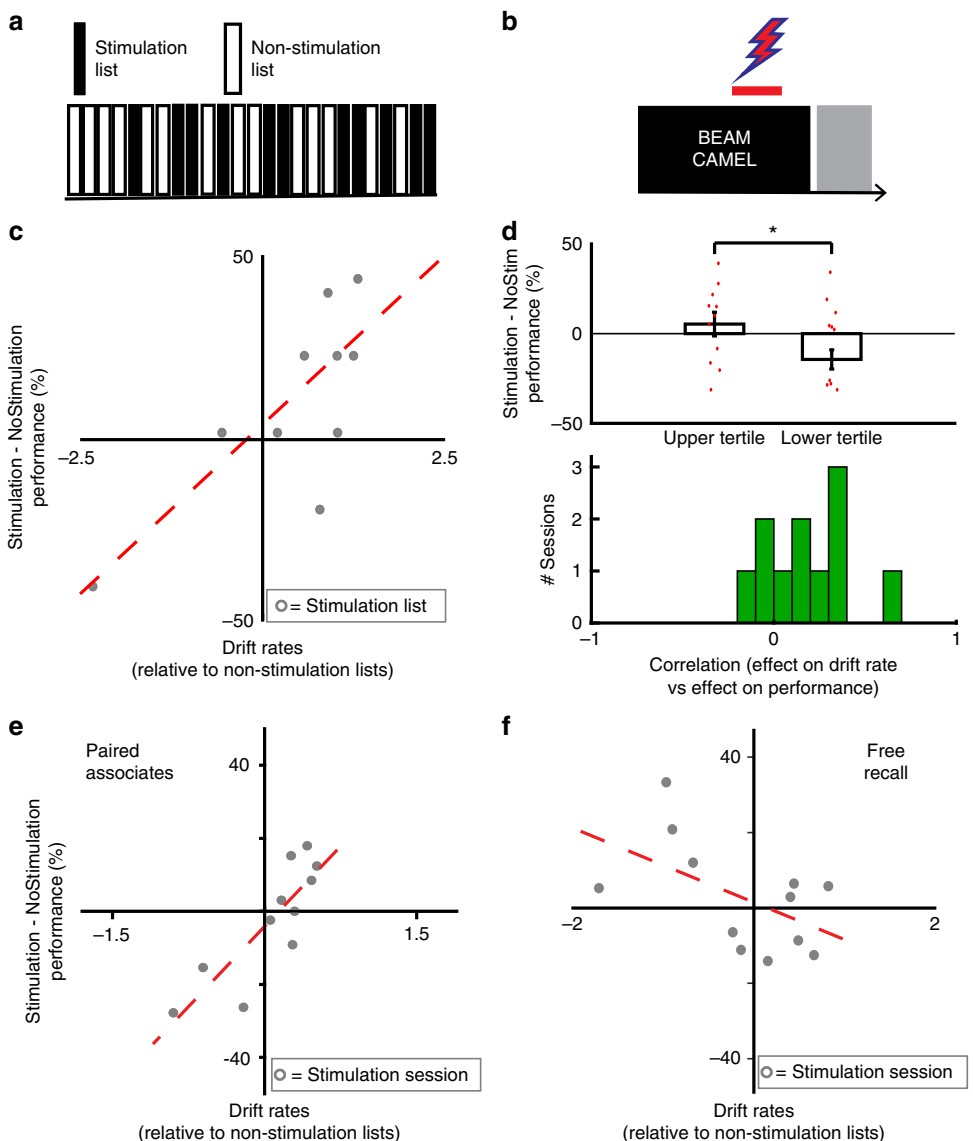

**Fig. 4** Electrical stimulation modulates rate of neural drift and memory performance. **a** For each session, 11 randomly chosen stimulation lists are interleaved with 11 non-stimulation lists. **b** During stimulation lists, electrical stimulation is applied during the encoding period. **c** Changes in the rate of neural drift due to stimulation and changes in memory performance due to stimulation, in a single exemplar participant in the paired associates task. Each data point represents a unique stimulation list. **d** Top, Change in memory performance in each experimental session between the highest and lowest tertile of lists, ranked by the changes in drift rate due to stimulation. Error bars represent SEM across 11 unique stimulation sites at which stimulation was applied. Bottom, Changes in the rate of neural drift due to stimulation and changes in memory performance in the paired associates task across experimental sessions. Each data point represents strength of correlation between stimulation-induced changes in drift rate and stimulation-induced changes in performance during an experimental session in which stimulation was applied at a unique pair of electrodes. **e** Average changes in the rate of neural drift due to stimulation and average changes in memory performance due to stimulation, in the paired associates task. Each data point represents a unique stimulation site at which stimulation was applied. **f** Average changes in the rate of neural drift due to stimulation and average changes in memory performance due to stimulation, in the free recall task. Each data point represents a unique stimulation site at which stimulation was applied

stimulation-induced increases in drift rate would be associated with stimulation-induced increases in performance, and vice versa. We correlated the effect of stimulation on performance with the effect of stimulation on drift rate across the 11 stimulation lists of each paired associates session (example; Fig. 4c). We found that there was a consistent correlation between effect of stimulation on performance and effect of stimulation on drift rate across participants (Fig. 4d, top; $r = 0.175 \pm 0.070$, $t(10) = 2.52$, $p = 0.030$, one-sample $t$-test). For each stimulation session, we then divided the stimulation lists into terciles based on the effect of stimulation on drift rate. We found that memory performance was significantly improved in lists in the upper tercile compared to the lower tercile across participants (Fig. 4d, bottom; $12.27 \pm 4.35\%$, $t(10) = 2.82$, $p = 0.018$, paired $t$-test).

We then examined whether the effect of stimulation on neural drift rate could explain the average effect of stimulation on performance, at the level of each session. By averaging the effect of stimulation across all stimulation lists in each session, and comparing this value to the average drift rate in the non-stimulation lists, we found that stimulation caused overall increases in the rate of neural drift in some participants, and overall decreases in others. The effects of stimulation on drift rate could not be explained by stimulation parameters or site of stimulation (Supplementary Fig. 6 and Supplementary Fig. 7). We examined whether the stimulation-induced changes in memory performance were correlated with stimulation-induced changes in rate of neural drift, across participants. To ensure that the resulting correlation was due specifically to the effect of stimulation, we used a permutation procedure that specifically compares the observed changes in drift rate and memory performance that result from stimulation to the relations between drift rate and memory performance that would arise by chance if stimulation had no effect (see Methods). Across participants, the average effect of stimulation on memory performance was significantly correlated with the average effect of stimulation on the rate of neural drift ($r = 0.78$, $p = 0.035$, permutation test; Fig. 4e). We did not find a significant relation between stimulation-induced changes in baseline similarity of neural activity and stimulation-induced changes in memory performance, suggesting that it is specifically the rate of neural drift which supports successful formation of distinct memories (Supplementary Fig. 8).

We conducted equivalent analyses on data from a separate cohort of participants who received stimulation while performing the free recall task ($N = 8$, 12 unique stimulation sites). We hypothesized that unlike in the paired associates task, any stimulation-induced increases in the rate of neural drift may result in worse rather than better memory performance. We found that the correlation between the effects of stimulation on drift rate and memory performance were significantly different in the two tasks (paired associates vs free recall, $F(1; 23) = 8.90$, $p = 0.0003$, one-way ANOVA; Fig. 4e, f), although the relation between effects of stimulation on drift rate and memory performance were not significant in free recall alone (free recall, $r = 0.46$, $p = 0.110$, permutation test; Fig. 4f). When we examined the effects of stimulation across lists, we did not find a significant relation between effect of stimulation on drift rate and effect of stimulation on performance in the free recall stimulation sessions (Supplementary Fig. 9). Finally, we confirmed that the decision to deliver closed-loop stimulation was not linked to changes in drift rate, by examining whether the output of the classifier was related to drift rate, and found no relations (Supplementary Fig. 10).

## Discussion
Retrieved context models provide a theoretical framework for how context shared across memories can promote the retrieval of multiple items[2,3,8,20,21]. Successful retrieval, however, also relies upon the ability to separate temporally adjacent memories from one another. Thus, shared contexts may be disadvantageous when the goal is to retrieve separate, individual episodes. We built upon this hypothesis by examining the extent to which overlap in temporal context during encoding is detrimental for memory retrieval in a paired associates verbal memory task, which requires the separation of individual memories. Our data demonstrate that selective retrieval of specific memories is at least partly facilitated by a higher rate of change in patterns of activity across the temporal lobe during encoding, which are likely to represent a more rapidly changing representation of temporal context.

We interpret our results in the framework of pattern separation models of memory which predict that reduced overlap between the neural representations of encoded items facilitates selective retrieval[12,13,22]. In these models, cues provided during retrieval function as inputs to an attractor network which ultimately succeeds or fails to complete a pattern of activity representing the original memory. The ability for such a network to successfully complete these patterns and retrieve a memory depends on how separable they are. Indeed, experimental evidence indicates that circuits in the medial temporal lobe are capable of orthogonalizing their inputs[15–17]. Furthermore, evidence indicates that the degree of drift in population activity over time is related to the degree of change in behavioral experience[11]. Our findings complement these results by suggesting that more orthogonal representations of temporal context, and therefore more separable patterns of activity, facilitate the selective retrieval of specific memories.

In contrast to the paired associates task, shared temporal contexts may be beneficial to the successful retrieval of multiple items in the free recall task[8,20,21]. Indeed, when we contrasted the relation between the rate of change in representations of temporal context and memory performance, we found significant differences between the two tasks. This difference was driven by the subset of free recall participants who demonstrated the strongest behavioral tendency to bind items across time. Hence, the degree to which the separation or binding of information across time is beneficial for memory likely depends upon the cognitive demands of the task at hand.

Direct cortical stimulation provided us with the opportunity to causally manipulate neural drift rates, and observe the concurrent effects on memory performance. Although various effects of electrical stimulation on memory have been reported, it is not well understood how these effects are related to changes in underlying neural activity[23–29]. Here, we found that stimulation caused increases in the rate of neural drift in some instances, and decreases in others. Importantly, we found that the effect of stimulation on memory performance was significantly correlated with the effect of stimulation on the rate of low frequency neural drift; as stimulation increased the rate of neural drift, memory performance in the paired associates task improved. These effects of stimulation were significantly different than those observed when stimulating during the free recall task. Altogether, these data provide further support to the hypothesis that selectively retrieving specific and distinct memories is facilitated when underlying representations of temporal context exhibit less overlap during encoding.

In our analyses, we examined the rate of change in a distributed pattern of spectral power across the temporal lobe. Notably, we found that the significant relationship between the rate of neural drift and memory performance was restricted to low frequency activity, suggesting that these frequencies may better represent temporal context. We also found that this effect emerges during interstimulus epochs, a result that is consistent with previous demonstrations that prestimulus low frequency activity predicts successful memory formation[30–32]. We interpret neural drift in distributed patterns of spectral power to be a reflection of

temporal autocorrelations in neuronal activity, of which examples have recently been reported[7,10,11]. Although previous investigations into the formation of distinct memory representations have focused on activity in regions of the medial temporal lobe, the fact that our results are driven by activity in lateral temporal cortex suggests that the rate of drift in cortical representations of temporal context contributes to the formation of distinct memories. The endogenous mechanisms responsible for maintaining and altering the rate of drift are unclear, and may include mechanisms such as those that govern the state of attention or arousal.

Taken together, our data bridge two parallel lines of research. On the one hand, temporal context models have found experimental support that shared context during encoding promotes the retrieval of temporally adjacent memories. On the other hand, computational models of memory suggest that retrieving individual memories relies on the ability to separate their neural representations. Our data are consistent with both proposals, as we demonstrate that separable representations of temporal context facilitate the selective retrieval of specific memories, thereby addressing the relatively unexplored question of how temporal context may be relevant when selectively retrieving specific memories. Moreover, our data suggest that any efforts to manipulate memory formation through electrical stimulation would benefit by accounting for and potentially targeting any endogenous processes that facilitate the encoding of distinct episodic memories[33].

## Methods

**Participants.** 86 participants with drug resistant epilepsy underwent a surgical procedure in which platinum recording contacts were implanted subdurally on the cortical surface as well as deep within the brain parenchyma. We initially started with a dataset of 33 participants who performed a paired associates task and 53 participants who performed a free recall task (Fig. 1a, b). We excluded data from two participants who performed the paired associates task because they did not complete a sufficient number of experimental lists to meet our inclusion criteria (see Behavioral Tasks). In addition, we excluded data from an additional three participants who performed the paired associates task and five participants who performed the free recall task because of insufficient electrode coverage in the temporal lobe of those participants (see Intracranial EEG Recordings). After excluding these participants from our initial dataset, we analyzed data from the 76 retained participants, 28 of whom performed the paired associates task and 48 of whom performed the free recall task.

For each participant, the clinical team determined the placement of the contacts as to best localize the epileptogenic focus. Data were collected at seven different hospitals: Clinical Center at the National Institutes of Health (NIH) (Bethesda, MD), the Hospital of the University of Pennsylvania (Philadelphia, PA), Thomas Jefferson University Hospital (Philadelphia, PA), Emory University Hospital (Atlanta, GA), Mayo Clinic (Rochester, MN), University of Texas Southwestern Medical Center (Dallas, Texas), and Dartmouth-Hitchcock Medical Center (Lebanon, NH). The research protocol was approved by the Institutional Review Board at each hospital, and informed consent was obtained from the participants and their guardians.

**Behavioral tasks.** Details regarding the paired associates and free recall tasks are provided in the Supplementary Information. For both tasks, we excluded all experimental sessions from our initial set of participants in which participants completed <20 lists (11 paired associates and 15 free recall sessions). This resulted in the complete exclusion of two participants who performed the paired associates task. The final dataset that we retained for our analyses of the experimental sessions that were performed without electrical stimulation included 28 participants who performed the paired associates task (63 experimental sessions, 2.33 ± 1.17 sessions per participant) and 48 participants who performed the free recall task (138 experimental sessions, 2.85 ± 1.61 sessions per participant). The final dataset that we retained for our analyses of the experimental sessions that were performed with stimulation included 7 participants who performed the paired associates task (11 unique stimulation sites, 1.18 ± 0.112 experimental sessions at each site) and 8 participants who performed the free recall task (12 unique stimulation sites, 1.75 ± 0.233 experimental sessions at each site).

**Intracranial EEG recordings and spectral power.** Whereas each hospital used the same general implantation procedures and data acquisition techniques, our analysis had to account for technical details that varied by institution. Depending on the amplifier, intracranial EEG (iEEG) acquisition system, and the discretion of the clinical team, iEEG signals were sampled at 500 or 1000 Hz. Signals were referenced to a common contact placed subcutaneously, on the scalp, or on the mastoid

process. All recorded traces were resampled at 1000 Hz, and a fourth-order 2 Hz stopband Butterworth notch filter was applied at 60 Hz to eliminate electrical line noise. The testing laptop sent either a 5 V analog pulses or a digital trigger via an optical isolator into a pair of open lines on the clinical recording system to synchronize the electrophysiological recordings with behavioral events. Subdural contacts were arranged in both grid and strip configurations with various inter-contact spacing. Depth electrodes were implanted in a subset of participants.

We analyzed iEEG data using bipolar referencing to reduce volume conduction and confounding interactions between adjacent electrodes[34]. We defined the bipolar montage in our dataset based on the geometry of iEEG electrode arrangements. For every grid, strip, and depth probe, we isolated all pairs of contacts that were positioned immediately adjacent to one another; bipolar signals were then found by differencing the signals between each pair of immediately adjacent contacts. The resulting bipolar signals were treated as new virtual electrodes (referred to as electrodes throughout the text), originating from the midpoint between each contact pair. All subsequent analyses were performed using these derived bipolar signals. In total, we utilized data from 8421 contacts localized to the temporal lobe (left hemisphere, 5032; right hemisphere, 3389). 1836 of these contacts were depth electrode recordings (left hemisphere, 1095; right hemisphere, 741). We recorded from 128.2 ± 6.57% electrodes for each participant in paired associates, and from 141.6 ± 5.60% electrodes for each participant in free recall.

To extract spectral power, we convolved the continuous time iEEG recording from each electrode with 43 logarithmically spaced complex valued Morlet wavelets (wave number = 5 cycles) ranging from 3 to 100 Hz. During the encoding period, we convolved each wavelet with the iEEG data from one second before the presentation of each item to stimulus offset (−1 s to 4 s for paired associates; −1 s to 1.6 s for free recall). We included a 1000 ms buffer on both sides of the clipped data. We squared and log-transformed the magnitude of the continuous-time wavelet transforms to generate a continuous measure of instantaneous power. To account for changes in power across experimental sessions, we z-transformed power values separately for each frequency and electrode using the mean and standard deviation of all 500 ms windows for that experimental session. We averaged the instantaneous power over each frequency band, then window of interest.

We minimized any confounding effects that may be related to transient epileptic activity (interictal discharges) by first removing all electrodes identified by clinicians as part of the seizure onset zone from further analysis. We then used an iterative cleaning procedure, in which we concatenated the raw voltage trace from all encoding epochs (−3000 ms to 6000 ms relative to the word pair presentation) separately for each electrode. We eliminated electrodes with kurtosis or variance greater than two standard deviations from the persistent sample mean. Based on these criteria, we excluded 27.4 ± 4.13% of electrodes in each participant. Following the exclusion of these electrodes, we then excluded all participants from our initial set of participants in whom less than ten electrode contacts were localized to the temporal lobe, in order to ensure that the distributed pattern of oscillatory power across this brain region was well represented. This resulted in the exclusion of three participants from our initial dataset who performed the paired associates task, and five participants who performed the free recall task. Finally, for each experimental session we then calculated the variance of the raw voltage trace for each encoding epoch, averaged across electrodes, and eliminated all trials with kurtosis or variance greater than two standard deviations from the persistent sample mean. Based on these criteria, we excluded 11.1±1.14% of trials in each free recall session and 13.4 ± 3.57% of trials in each paired associates session.

**Anatomical localization.** To localize electrode contacts, we first constructed a cortical surface for each participant using the pre-implant whole brain volumetric T1-weighted MRI scans (Freesurfer)[35]. We established coordinates for the radio-dense electrode contacts using a post-implant computed tomography scan, and then registered the CT scan with the pre-operative MRI using Advanced Normalization Tools (ANTS)[36]. Subdural electrode coordinates were further mapped to the cortical surfaces using an energy minimization algorithm[37]. Two neuroradiologists reviewed cross-sectional images and surface renderings to confirm the output of the automated localization pipeline. For our analyses, we defined the lateral temporal lobe to consist of electrodes localized to inferior, middle, or superior temporal gyrus, and the medial temporal lobe to consist of electrodes localized to the parahippocampal gyrus, perirhinal cortex, entorhinal cortex, hippocampus, or amygdala.

**Neural drift.** We constructed a feature vector associated with each item j using the average z-scored power of each frequency in each electrode localized to the temporal lobe:

$$\vec{E}_j = \left[ z_{1,1}(j) \dots z_{1,F}(j) \dots Z_{L,F}(j) \right]$$

where $z_{l,f}(j)$ is the z-transformed power of electrode $l = 1 \dots L$ at frequency $f = 1 \dots F$ in the temporal lobe averaged over the time period of interest relative to the presentation of each word or word pair $j$.

In our initial analysis of overall levels of neural drift, we used the power averaged over the interstimulus period (−750 to 0 ms relative to the presentation of each item) to calculate the average z-transformed power for each frequency, and then constructed a feature vector for each item by combining values of power

across five frequency bands (Fig. 1): theta (3–8 Hz), alpha (8–12 Hz), beta (13–25 Hz), low gamma (30–58 Hz), and high gamma (62–100 Hz)[9]. Then, in order to determine whether the relation between neural drift rate and memory performance was specific to individual frequencies and time points, we separately constructed feature vectors using each combination of individual frequencies and sliding 500 ms window (100 ms steps, 80% overlap; Fig. 2b). In post-hoc and stimulation analyses, we specifically focused on low frequency activity in the interstimulus epoch, and therefore constructed feature vectors using the z-transformed power averaged over the entire interstimulus epoch (−750 to 0 ms) and over all frequencies between 3 and 12 Hz.

To assess the extent to which the distributed pattern of spectral power changes over time, we computed the cosine similarity between feature vectors assigned to every epoch, $E_j$ and every other epoch within the same list, $E_{j+n}$, where $n$ is the number of intervening items, or lags. Cosine similarity provides a measure of how similar the angles of two vectors are in a multidimensional space, and in this case reflects how similar the neural pattern of activity is between any two time points. Within each list of length k items, we therefore computed the average similarity between feature vectors separated by identical time lags, $n$:

$$p_n = \frac{1}{k-n}\sum_{j=1}^{k-n} \frac{\vec{E_j} \cdot \vec{E_{j+n}}}{\left\|\vec{E_j}\right\|\left\|\vec{E_{j+n}}\right\|} \qquad (1)$$

where $_n$ corresponds to the average similarity of epochs spaced apart by n lags in a given list. In a list of $k$ items, the number of comparisons contributing to the average similarity between epochs separated by only one lag is k − 1, whereas the similarity between epochs separated by k − 1 lags is made of one comparison.

We defined the rate of neural drift within each list as the extent to which epochs separated by only one lag were more similar than epochs separated by two lags:

$$\frac{p_1 p_2}{p_1} \qquad (2)$$

Neural drift rate therefore measures how quickly the distributed pattern of spectral power in the temporal lobe changes over time. We used the similarity of epochs separated by one or two lags to generate our measure of neural drift rate because the average similarity at these lags describes the greatest number of observations, and is therefore the most reliable measure of change. Within each experimental session, we calculated the Spearman correlation between the neural drift rate and performance in each list. For participants who completed more than one session, we defined the participant level correlation as the average of the correlations across sessions.

**Temporal clustering**. For each participant, we derived a value of temporal factor, which reflects the degree to which recalled items tend to be linked to one another in time, or clustered[21]. For each behavioral session, we computed this value by first labeling all correctly recalled items with the serial position in which they were encoded. For each recalled item, we then calculated the transition distance, or distance from the previously recalled item in units of serial position. All possible distances are ranked in order of the negative absolute value of their transition distances. The temporal factor for each recall is then computed as: (R−1)(N−1), where R is the rank of the actual recall and N is the number of possible recalls. Each transition receives a number between 0.0 and 1.0, where factors greater than 0.5 indicate that the participant selected a temporally adjacent word, and factors <0.5 indicate that the participant selected a temporally distant word, relative to all possible valid transitions[21].

**Statistical analysis**. Data are presented as mean±SEM. Unless otherwise specified, all statistical comparisons were conducted as two-tailed tests. We utilized Spearman's rank correlation when evaluating the monotonic relationship between two variables. Spearman's correlation utilizes only the order of data points and is thus not biased by outliers as with Pearson's correlation. To compare correlations between paired associates and free recall, we used a Fisher z-transformation on the correlation coefficients.

The transformation stabilizes the variance of these correlations, reduces bias towards lower correlations, and results in a normalized distribution of coefficients. For each correlation, we therefore calculated the Fisher z-transform: $z = \frac{1}{2}\ln\frac{1+r}{1-r}$ where $r$ is the correlation coefficient. For participants who completed more than one session, we applied the z-transform after calculating the average correlation coefficient across sessions. To identify specific time points and frequencies that exhibit a significant correlation between neural drift rate and memory performance, we separately constructed feature vectors for every individual frequency and time window, to generate a measure of neural drift rate in each list. We then correlated neural drift rate with memory performance, to obtain a matrix of correlation coefficients in each participant, representing every individual frequency and time window. For each individual time-frequency combination, we then tested whether the distribution of correlation coefficients across participants was significantly different from zero using a one-sample $t$-test. This generates a t-statistic and p value for every individual time-frequency combination, but does not correct for the multiple comparisons that are made across all frequencies and time points.

To address this, we used a nonparametric clustering-based procedure[38]. We first identified contiguous clusters of at least ten time-frequency points that each exhibit a significant correlation between rate of neural drift and memory performance across participants ($p < 0.05$). For each identified cluster, we computed a cluster statistic by taking the sum of the t-statistic across all time-frequency points contributing to that cluster. We then performed a permutation procedure. In each of 1000 permutations, we randomly flipped the sign of each correlation coefficient in each time-frequency point in each participant. In each permutation, we then tested the distribution of randomly inverted correlated coefficients across participants for each time-frequency point. Once again, we identified clusters of time and frequency that exhibited a distribution of coefficients that were each significantly different from zero, and calculated a cluster statistic for each of the identified clusters in each permutation. We used the maximum cluster statistic from each permutation, and in this manner generated an empiric distribution of 1000 maximum cluster statistics that would arise by chance. The distributions were visually inspected for normality. To generate a p value for each cluster in the true dataset, we compared the position of the true cluster statistic to the distribution of maximum cluster statistics from the permuted cases. Clusters were deemed significant if $p < 0.05$.

**Closed-loop stimulation**. The closed-loop stimulation sessions consisted of four practice lists followed by 22 task lists. We applied electrical stimulation during the encoding period in 11 of the lists, chosen at random. Prior to the stimulation sessions, we trained an L2-regularized logistic regression classifier using data captured during previous sessions to produce a set of weights mapping spectral features of iEEG activity to an output probability of later word recall. During the 11 stimulation lists, we applied the classifier weights to the pattern of spectral power corresponding to the encoding of each item. Stimulation was applied only when the predicted probability of recall fell below the median threshold, such that approximately half of the items on each list were stimulated (paired associates, 0.45 ± .033; free recall, 0.47 ± .021). Details are provided in the Supplementary Methods and elsewhere[27].

In order to assess the effects of stimulation on neural drift, for each list we computed the rate of neural drift using z-transformed power averaged over the entire interstimulus epoch (−750 to 0 ms) and over all frequencies between 3 and 12 Hz. This is the same time period and range of frequencies identified as exhibiting a significant relation between neural drift rate and memory performance in the sessions without stimulation and used for subsequent post hoc analyses. When constructing the feature vectors to compute the neural drift rate in the free recall data, we excluded feature vectors from interstimulus epochs which followed stimulation epochs, to avoid including artifactual data in our analyses. For paired associates stimulation sessions, there was no overlap between interstimulus intervals and stimulation epochs, thus no feature vectors were excluded. For each session, we analyzed the rates of neural drift during the 11 non-stimulation lists to derive a distribution of baseline rates of neural drift. We then z-scored the neural drift rate in each of 11 stimulation lists using the mean and standard deviation of the non-stimulation distribution. For each experimental session, we calculated the average of these z-scored values across stimulation lists to get a single metric for the change in neural drift due to stimulation in each session.

To assess the effects of stimulation on performance, for each experimental session we calculated the difference in successful recall percentage during all stimulation lists compared to non-stimulation lists. To account for baseline performance levels, we divided this difference by the total recall percentage in the entire session. The resulting value reflects the degree to which electrical stimulation during that session improved or worsened memory performance when compared to lists without stimulation. For participants who completed stimulation sessions at different electrode pairs, we considered each session as an independent experimental session. If multiple sessions were conducted at the same electrode pair, we averaged the values described above.

We examined the relation between changes in the rate of neural drift and changes in memory performance across participants using Spearman's correlation. We used a permutation test to determine if the observed correlation was significantly different than expected by chance, and thus due specifically to the effect of stimulation. For each permutation, we randomly shuffled the labels for stimulation and non-stimulation lists, and recomputed the changes in neural drift rate and performance. This procedure, conducted 1000 times, generates an empiric distribution of 1000 null correlation values. We compared the Spearman's correlation ($r$) calculated using the true data to this distribution of permuted data, and assigned a p value based on the rank of the true value in this permuted distribution.

**Reporting summary**. Further information on experimental design is available in the Nature Research Reporting Summary linked to this article.

## Data availability

Processed data used in this study can be found at: https://neuroscience.nih.gov/ninds/zaghloul/downloads.html. Custom MATLAB analysis code is available upon request.

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

## Acknowledgements

This work was supported by the Intramural Research Program of the National Institute for Neurological Disorders and Stroke and partially supported by the DARPA Restoring Active Memory (RAM) program (Cooperative Agreement N66001-14-2-4032). We thank Medtronic and Blackrock Microsystems for providing neural recording and stimulation equipment, Michael J. Kahana for providing helpful feedback and access to the free recall dataset, and clinical teams at the various hospitals for collection of data. We are indebted to all patients who have selflessly volunteered their time to participate in this study. The views, opinions, and/or findings contained in this material are those of the authors and should not be interpreted as representing the official views or policies of the Department of Defense or the U.S. Government.

## Author contributions

M.M.E., J.J.W., T.C.S. and S.K.I. performed data collection; M.M.E., J.J.W., V.S. and K.A.Z. designed analyses; M.M.E. analyzed data; M.M.E. and K.A.Z. wrote the manuscript. All authors provided feedback on the manuscript. K.A.Z. supervised the research.

## Additional information

**Competing interests:** The authors declare no competing interests.

