## [Peer Review File · Nature Communications]

Reviewers' comments:

Reviewer #1 (Remarks to the Author):

In this paper, the authors explore the hypothesis that temporal context (as measured by smoothly changing patterns of neural activity) facilitates some kinds of episodic memories, whereas others it would impair. This hypothesis is very well motivated in the experimental (including several intracranial studies) and theoretical literature and the two experiments done here are very well suited to address both aspects of the hypothesis. The writing is mostly clear and the paper is easy to understand (with the exception of some statistical issues). The hypothesis explored and results found are novel and interesting. Of particular note is the tests of causality using electrical stimulation.

The analysis approach of using the data from inter-stimulus intervals and with bipolar recordings is appropriate and well thought out. The datasets used are very large and unique, providing incredible access directly to the human brain.

Major claims are: i) the larger the drift rate, the better is performance in paired associate learning (a task where context would be expected to impair performance), ii) for patients which took most advantage of clustering in free recall, this same relationship was the opposite: slower drift predicts good performance, iii) stimulation during paired associate task either increased or decreased drift, and this either impaired or improved performance, respectively. Weaknesses of the approach and statistical issues are summarized below. In summary I believe this is an important contribution to the literature, and given proper attention to below issues suitable for publication.

Major issues:

1. Anatomy. No information is given on where the electrodes recorded are (beyond that they are in the 'temporal lobe'), nor which of these contribute to the contiguity effect. A figure should be added showing localization data and analysis performed that shows which area(s) contribute to these effects. I realize the metric used is a population metric across all electrodes of a patient, but nevertheless the question applies.

2. The patients that performed the two tasks were disjoint, with no overlap (between subject design). This raises some concerns that perhaps the different effects are due to differences in the patients or electrode placement. The similar decay in pattern similarity argues against this neurally. Nevertheless, the absence of any other information to assess how well matched these two groups were is not helpful and should be provided.

3. Stimulation result. While principally a very interesting result, the analysis has several caveats and requires substantially more detail to be useful. A principle problem is the margin significance of the stimulation results.

3.1. It is stated that 'in some patients (stimulation sites), stimulation increased drift; in others, it decreased drift'. This is a potentially powerful link between the contradictory results of stimulation that predicts beneficial effects in some tasks but not others. This would predict that the same stimulation improves one task but disrupts the other and vice-versa, but if I understand the paper correct the two stim patient groups were disjoint? (clarify). Also, if drift is truly predictive of performance, one would like to see more than a correlation: rather, split sessions into two groups (drift increase, drift decrease). Then, for the individual groups, was performance significantly improved/disrupted vs. no performance difference?

3.2. The principle parameter here is a change in drift speed. But did stimulation also affect baseline similarity or was it only the speed of change? (similar to what is addressed in S4)

3.3. stimulation in free recall had no significant effect (Fig. S5), but in the manuscript the statement is made that "that they were in the opposite directions in the two tasks". This is not a

valid statement.

3.4. One would like to understand why some stimulation sites/parameters cause an increase in drift, whereas an other cause a decrease. Was there something systematically different between stim sites, parameters, triggering protocol?

3.5. What is the purpose of the closed-loop stimulation? Specifically of relevance here, did, at the points of time when the decoder decided whether to stimulate or not, the drift rates differ? Since the stimulator already predicted 'bad performance' when it was turned on, might this be what triggered the stimulation?

4. The manuscript is unnecessarily short, with many essential analysis and figures moved to supplement. It is unclear to me why that is. For example, S3 should clearly be part of Fig 3, otherwise the not sig different from zero effect for free recall is confusing. Same for S4, a needed analysis. All this needs to be integrated into one complete manuscript of the kind expected for this journal.

5. Statistics. There are several instances in the manuscript of "is of the opposite sign" when it is in fact not significantly different from zero. That is not a statistically valid conclusion. This concerns in particular the correlation of performance with drift in the free recall task, which on average is not significantly different from zero (except in S3 when using the patients that cluster well). This also applies to stimulation during free recall. There are of course statistical tests for sign changes, but the authors don't use these.

Reviewer #2 (Remarks to the Author):

The paper "Changing temporal context in human temporal lobe promotes memory of distinct episodes" by El-Kalliny et al. reports on intracranial EEG recordings from the temporal lobe in 78 epilepsy patients performing a memory task involving either free recall of a learned word list or, alternatively, cued recall from a learned list of word pairs. Focusing on the role of context during memory encoding, the authors quantify the amount change in power-spectral similarity from one trial lag to two lags as neural drift rate. Hypothesizing that changing context as measured by neural drift could be beneficial for separating distinct memories such as word pairs, but detrimental for associating sequential items in a word list, they indeed find the neural drift rate during encoding to be positively correlated with retrieval performance in the paired-associate task, but negatively correlated during the free recall task. In addition, they perform intracranial electrical stimulation in some patients. The resulting changes in neural drift across patients include both increase and decrease, but generally confirm the expected effects on the different memory tasks. The authors argue that their findings reconcile theories on temporal context as a binding feature and computational models on the role of temporal context in pattern separation.

Intracranial in-vivo recordings from the human brain are rare and precious, and the extraordinarily large number of subjects included in this study can only be obtained in a multi-center effort, which makes it even more special. The methods used in the study appear to be sound, although little bits of information are missing here and there. The paper is very well written and comes from a group well established in the field of human intracranial recordings. There is a lot to be liked about this paper. Nonetheless, I have a few concerns that should be addressed before publication.

Major points.

(1) Some important information is missing in the paper. For instance, we are never explicitly told how many patients were eventually included in the analyses. Buried deep in the Supplement, we learn that subjects "in whom less than 10 electrode contacts were localized to the temporal lobe" were excluded from the analysis, but we are not told how many. Judging from the degrees of freedoms indicated in the t-test results ($t(74)$ and $t(27)$), 2 patients from the word-pair group

seem to have been excluded, leaving 28 vs. 48 in both groups. However, statistical test descriptions in the SI Results ($t(48)$, $t(28)$, $t(76)$) are inconsistent with this. In the caption of Fig. 1, the authors mention "28 participants who performed the paired associates task, and 47 participants who performed the free recall task", which again is inconsistent. The correct and final number of subjects included in the analysis should be made explicit right from the beginning. The authors may mention in the SI that the original group size was even larger, but subjects had to be excluded for certain reasons. Reading about 78 subjects and finding out much later that only 76 or 75 of them were included in the relevant analyses is unnecessarily misleading.

Also, we never learn how many sessions were actually recorded and analyzed. The total number, mean, and range across patients should be given, separately for sessions with and without stimulation, so we know exactly how many sessions were included in which analysis. When mentioning in the SI that "all experimental sessions in which a participant completed less than 20 lists" were excluded, the authors should state how many these were.

On a similar note, instead of mentioning task descriptions like "up to 25 lists", "up to 150 word pairs", or "up to 300 total words", it would be more informative to give the mean and range across sessions, again separately for sessions with and without stimulation.

Finally, for the number of included electrode contacts, stating the range across patients would be informative.

(2) The results from the electrical stimulation paradigm are quite intriguing. It would strengthen the authors conclusions if they could explicitly show that changes in neural drift rate caused by electrical stimulation explain changes in memory performance better than the presence of electrical stimulation per se.

Minor points.

(1) In the Results section the authors state "The extent to which any pair of epochs had similar activity significantly decreased as more time elapsed between them" and then state results from an ANOVA to statistically undermine their statement. However, an ANOVA does not test for an increase or decrease or any directed relationship. Please use an appropriate statistical test (e.g., Spearman correlation).

(2) It is not entirely clear to me whether the analyses related to Figs. 1 and 2 included exclusively "record-only sessions". Perhaps this could be made clearer.

(3) For the data shown in Fig. 1E and related analyses, it is not clear to me over which time and frequency range the data was averaged. Please clarify.

(4) p. 8/9: "we [...] constructed feature vectors using the z-transformed power averaged over the interstimulus epoch (-750 to 0 ms) and over 3 to 12 Hz" Was this entire range used or only time-frequency windows belonging to the significant cluster in Fig. 2B?

(5) p.10: "we computed the rate of neural drift in the activity of interest." What was this activity of interest? The same range as in point (4)?

(6) Related to the previous 3 points, the authors should go over their paper and make sure that for each and every neural drift rate calculated, it is absolutely clear from which time-frequency range it was derived.

(7) For the results shown in Figs. 3C, S3, S4, results of the one-sample t-test against 0 should also be given in cases where they were not significant in order to convey a feeling of whether there was a statistical trend or not.

(8) p.9: "For participants who completed more than one session, we defined the participant level

correlation as the average of the correlations across sessions." This sentence comes before the first mention of the Fisher-z transform. Did the authors use the transform for this average also?

(9) p.10: "In each of 1000 permutations, we randomly inverted the sign of the correlation coefficient in each time-frequency point in each participant." Was this a random permutation of existing signs or a random recreation of each single sign?

(10) In the SI, the authors describe the Morlet wavelets they used once as "wavelet number = 5" and once as "wave number = 5". Is this number the "number of cycles"?

(11) SI Anatomical Localization: A common electrode target in the temporal lobe is the amygdala. Were there electrode contacts in the amygdala, and if so, were they excluded for a particular reason?

(12) SI Closed Loop Electrical Stimulation: The authors state that to determine a safe stimulation amplitude, they went up to 1.5 mA for depth contacts. A few sentences later they state that in the paradigm they used up to 2 mA at depth contacts. Is this correct?

(13) SI Closed Loop Electrical Stimulation: "Data captured during non-stimulation lists were also used for the z-transform of input features." Why "also"?

(14) SI Closed Loop Electrical Stimulation: "In cases in which no lateral temporal cortex contacts were available, we selected an electrode pair at or near the largest SME elsewhere in the brain." How many cases?

(15) SI Closed Loop Electrical Stimulation: "For a subset of sessions, the particular amplitude and frequency used for stimulation were chosen based on a pre-test in which we stimulated the brain at each parameter combination, while the patient was at rest (no experimental task)." How many sessions?

(16) SI Results: The text referring to Fig. S1 is a bit confusing. Maybe switching panels A and B would help.

Typos.

-SI, p.2: "Based on this criteria" should be "Based on this criterion" or "Based on these criteria".

-SI, p.3: "we trained a logistic regression classifier trained to ..." Omit the second "trained".

Reviewer #3 (Remarks to the Author):

This manuscript addresses the question whether changes in temporal context in terms of spectral EEG patterns within the temporal lobe influence verbal memory performance. The analyses are based on intracranial EEG recordings in 78 presurgical epilepsy patients, performing either a paired associates task or a free recall task. The authors hypothesized that decreased spectral similarity of inter-stimulus EEG during encoding is related to increased performance in the paired associates task, because spectral similarity may impede pattern separation. On the other hand, they supposed that free recall performance may rather be facilitated by context similarity. As major results the authors report that the rate of drift of low frequency EEG patterns within temporal lobe indeed is positively correlated with performance in the paired associates task. This relation is found to be opposite for the free recall task. Finally, the authors show that context similarity can be modulated by electric stimulation and that stimulation-related changes in context similarity

correspond to changes in memory performance.

In my view, the questions and hypotheses underlying this study are very timely and important for the understanding of human memory dynamics. The manuscript is based on a precious data set, i.e. a large number of intracranial EEG recordings in epilepsy patients performing verbal memory tasks. The analysis approaches and techniques appear highly sophisticated and up-to-date. Importantly, the electric stimulation sessions provide additional information regarding the causal relevance of the observed interrelations between context similarity and memory performance. The interpretation and discussion of the results appears straightforward and convincing. Thus, I have only a few comments, which are listed below:

Major:

- The authors present behavioral evidence for temporal contiguity effects in the free recall data (supplement). It should be considered whether also a measure reflecting the degree of cognitive separation in the paired associates task could be calculated, for instance, by evaluating intrusions (i.e. wrong associations belonging to other word cues). Similar to the results for the free recall task, this measure then could be related to the observed correlations between context similarity in terms of iEEG patterns and memory performance.
- The authors implemented artifact rejection only in terms of a channel exclusion procedure. There still may be a considerable amount of trials contaminated by epileptiform activity in the remaining channels. Since proper artifact rejection is essential, in particular, when dealing with iEEG data from epilepsy patients, the authors should provide evidence that their results are not affected by epileptiform activity.
- Do the authors observe any laterality effects, i.e. do the results differ when being separately evaluated for the left and the right hemisphere?

Minor:

- Page 4: "We found that the rate of drift ... was correlated ... across all participants": please state the average correlation coefficient across subjects.
- Page 5: "... the decision to stimulate was controlled by a classifier ...". Which classifier features predicted successful memory retrieval?
- Figure 1c: The timing of stimulus presentations during the encoding phase is quite different for the two paradigms. Please discuss whether or not this may have had an impact on the results.
- Supplement, page 5: "We found no significant clusters ...". In my view, this result regarding the relation between iEEG pattern drift and free recall performance should be reported in the main text.
- Supplement, page 5: "The tertile of participants ... demonstrated a relation ...". Again, in my view, this result should be reported in the main text.
- Supplement, page 5: "We divided the participants ... averaged across all recalls (0.664 +/- 0.0078). Please add statistics testing whether this temporal contiguity value is above chance (i.e. above 0.5).

Reviewer #1 (Remarks to the Author):

In this paper, the authors explore the hypothesis that temporal context (as measured by smoothly changing patterns of neural activity) facilitates some kinds of episodic memories, whereas others it would impair. This hypothesis is very well motivated in the experimental (including several intracranial studies) and theoretical literature and the two experiments done here are very well suited to address both aspects of the hypothesis. The writing is mostly clear and the paper is easy to understand (with the exception of some statistical issues). The hypothesis explored and results found are novel and interesting. Of particular note is the tests of causality using electrical stimulation.

The analysis approach of using the data from inter-stimulus intervals and with bipolar recordings is appropriate and well thought out. The datasets used are very large and unique, providing incredible access directly to the human brain.

Major claims are: i) the larger the drift rate, the better is performance in paired associate learning (a task where context would be expected to impair performance), ii) for patients which took most advantage of clustering in free recall, this same relationship was the opposite: slower drift predicts good performance, iii) stimulation during paired associate task either increased or decreased drift, and this either impaired or improved performance, respectively. Weaknesses of the approach and statistical issues are summarized below. In summary I believe this is an important contribution to the literature, and given proper attention to below issues suitable for publication.

We thank the Reviewer for these encouraging comments.

Major issues:

1. Anatomy. No information is given on where the electrodes recorded are (beyond that they are in the ‘temporal lobe’), nor which of these contribute to the contiguity effect. A figure should be added showing localization data and analysis performed that shows which area(s) contribute to these effects. I realize the metric used is a population metric across all electrodes of a patient, but nevertheless the question applies.

We thank the Reviewer for this suggestion. We have now added a supplementary figure (Supplementary Fig. 1) illustrating the electrode coverage in the two tasks from all the participants. In addition, we have performed an additional analysis examining which areas contribute to the effect we report. We found the relation between neural drift and memory performance is driven by activity in lateral temporal cortex, and not in medial temporal lobe. We did not find evidence of this effect either globally, or in other brain regions such as the frontal or parietal lobe. We have added the results of these new analyses to our revised Results. In the supplementary figure showing the electrode coverage, we have also demonstrated the regional division of the electrodes used for these analyses.

Results

Our data examining the relation between drift rate and memory performance in the two tasks demonstrate that in paired associates, faster rates of neural drift in the temporal lobe specifically support the ability to retrieve separate distinct items. We were interested, however, in whether the rate of neural drift or the degree of baseline similarity may be a global phenomenon across the entire brain, or specific to individual brain regions. As such, we repeated our analyses, in this case using the distributed pattern of 3-12 Hz power during the interstimulus period across all brain electrodes in each participant. We found no significant relation between rate of neural drift and memory performance in either task when we use a whole brain representation of temporal context (paired associates, $t(27) = 0.339$, $p = .74$, free recall, $t(47) = -1.19$, $p = .24$). Similarly, we found no significant relation between baseline similarity, computed using the whole brain representation, and memory performance in either task (paired associates, $t(27) = -1.44$, $p = .0.161$, free recall, $t(47) = -1.48$, $p = .0.149$). We examined the relation between neural drift and memory performance selectively using only patterns of activity in the frontal lobe or the parietal lobe, and also found no significant relation across participants in either task (frontal lobe; paired associates, $t(21) = 0.652$, $p = .520$, free recall, $t(33) = 1.19$, $p = 0.244$; parietal lobe paired associates, $t(17) = 0.364$, $p = .712$, free recall, $t(28) = -0.480$, $p = .0.636$).

Conversely, although our data suggest that overall temporal lobe representations of temporal context underlie these phenomena, it is possible that the relation between neural drift and memory performance may localize to subregions of the temporal lobe. Given previous evidence that the medial temporal lobe captures representations of a memory’s spatiotemporal context (Folkerts et al., 2018; Tsao et al., 2018), we hypothesized that the differing pattern of results in paired associates and free recall would be driven by activity of the medial temporal lobe. We therefore conducted separate analyses examining the representations of neural drift in the medial and lateral temporal lobe and their relation with memory performance. Surprisingly, we

found that the relation between memory performance in the paired associates task and the rate of neural drift, and the difference in these relations between the two tasks, was significant for neural activity in the lateral temporal cortex (paired associates, $t(26) = 3.20$, $p = .0035$; free recall, $t(46) = -1.45$, $p = .15$; paired associates vs free recall, $t(73) = 3.17$, $p = .002$), but not for neural activity in the medial temporal lobe (paired associates, $t(14) = 1.08$, $p = .29$, free recall; $t(20) = -0.513$, $p = .61$; paired associates vs free recall, $t(35) = 0.449$, $p = .66$).

2. The patients that performed the two tasks were disjoint, with no overlap (between subject design). This raises some concerns that perhaps the different effects are due to differences in the patients or electrode placement. The similar decay in pattern similarity argues against this neurally. Nevertheless, the absence of any other information to assess how well matched these two groups were is not helpful and should be provided.

The Reviewer raises an important concern. We believe that the two cohorts of participants are indeed well matched for the following reasons. First, as the Reviewer notes, both sets of participants exhibit similar rates of neural drift. Second, both sets of participants have similar electrode coverage, suggesting that neural activity used in our analysis is captured from similar brain regions. Third, both sets of participants have similar ages and levels of IQ. Fourth, we have added an additional analysis in which we examined the changes in neural activity during the encoding period that occur with successful memory formation (subsequent memory effect), and have found that both sets of participants exhibit similar changes. We now report these similarities in the revised Results, and have added a new Supplementary Figure (Figure S3) to illustrate the similarity in the subsequent memory effect between the two groups.

Results

Both sets of participants that performed the paired associates task and the free recall task exhibited similar rates of neural drift, similar electrode coverage (Supplementary Fig. 1), similar ages (paired associates, age 33.8 ± 1.35 years; free recall, age 36.9 ± 1.62 years), similar IQ levels (paired associates, WAIS IV FSIQ 89.5 ± 4.39 ; free recall, WAIS IV FSIQ 88.6 ± 2.81), and exhibited similar changes in low and high frequency power in the temporal lobe during successful encoding (subsequent memory effect; Supplementary Fig. 3), suggesting that both sets of participants are well matched with respect to the neural processes that underlie memory formation.

3. Stimulation result. While principally a very interesting result, the analysis has several caveats and requires substantially more detail to be useful. A principle problem is the margin significance of the stimulation results.

3.1. It is stated that ‘in some patients (stimulation sites), stimulation increased drift; in others, it decreased drift’. This is a potentially powerful link between the contradictory results of stimulation that predicts beneficial effects in some tasks but not others. This would predict that the same stimulation improves one task but disrupts the other and vice-versa, but if I

understand the paper correct the two stim patient groups were disjoint? (clarify).

We agree with the Reviewer that a potentially powerful analysis would be to examine the effects of stimulation in the same participant as they perform the two separate tasks. Unfortunately, as the Reviewer has correctly noted, we performed the analysis on the effects of stimulation during the free recall task on a separate set of participants from those who received stimulation during the paired associates task. We did not have any participants in our data set who performed a stimulation experiment with both tasks. We have clarified this point in the text.

Results

We conducted equivalent analyses on data from a separate cohort of participants who received stimulation while performing the free recall task.

Also, if drift is truly predictive of performance, one would like to see more than a correlation: rather, split sessions into two groups (drift increase, drift decrease). Then, for the individual groups, was performance significantly improved/disrupted vs. no performance difference?

We think this is a good suggestion. We have now conducted two additional analyses to more explicitly examine the effects of stimulation on drift rate and memory performance. First, in each participant, we examined the effect of stimulation on drift rate and performance within each list of the experimental session (previously, we had examined the overall effects on the aggregate performance across all lists for each participant). Across participants, we found a consistent correlation between the effect of stimulation on performance and the effect on drift rate. Second, as the Reviewer suggested, we then split the lists in each participant into terciles, based on the effect of stimulation on drift rate. We compared memory performance between the lists from the upper tercile to the lower tercile, and found across participants a significant improvement in memory in the upper tercile of lists, marked by the faster drift rates. We have now described these additional results in the revised Results.

Results

Given the results observed during passive recordings of iEEG activity, we hypothesized that stimulation-induced increases in drift rate would be associated with stimulation-induced increases in performance, and vice versa. We correlated the effect of stimulation on performance with the effect of stimulation on drift rate across the 11 stimulation lists of each paired associates session (example; Fig. 4C). We found that there was a consistent correlation between effect of stimulation on performance and effect of stimulation on drift rate across participants (Fig. 4D, top; $r = 0.175 \pm 0.070$, $t(10) = 2.52$, $p = 0.030$). For each stimulation session, we then divided the stimulation lists into terciles based on the effect of stimulation on drift rate. We found that memory performance was significantly improved in lists in the upper tercile compared to the lower tercile across participants (Fig. 4D, bottom; $12.27 \pm 4.35\%$, $t(10) = 2.82$, $p = 0.018$).

3.2. The principle parameter here is a change in drift speed. But did stimulation also affect baseline similarity or was it only the speed of change? (similar to what is addressed in S4)

We examined the effects of stimulation on baseline similarity as well, and found that the effect of stimulation on performance was not correlated with the effect of stimulation on baseline similarity. We have presented this result in the revised Results, and these data are now also presented in Supplementary Fig. 8.

Results

We did not find a significant relation between stimulation-induced changes in baseline similarity of neural activity and stimulation-induced changes in memory performance, suggesting that it is specifically the rate of neural drift which supports successful formation of distinct memories (Supplementary Fig. 8).

3.3. stimulation in free recall had no significant effect (Fig. S5), but in the manuscript the statement is made that “that they were in the opposite directions in the two tasks”. This is not a valid statement.

We agree with the Reviewer that the claim that the results of the analysis on the effects of stimulation on free recall memory performance were in the opposite direction is not valid. We have removed all instances in which this wording was used, and have instead focused on the difference in the effects of stimulation on paired associates compared to free recall.

Results

Indeed, we found that the correlation between the effects of stimulation on drift rate and memory performance were significantly different in the two tasks, although the relation between effects of stimulation on neural drift and memory performance were not significant in free recall alone (paired associates vs free recall, $F(1, 23) = 8.90$, $p = .0003$, one-way ANOVA; free recall, $r = -0.46$, $p = .110$, permutation test; Fig. 4E,F).

3.4. One would like to understand why some stimulation sites/parameters cause an increase in drift, whereas another cause a decrease. Was there something systematically different between stim sites, parameters, triggering protocol?

We agree with the Reviewer that this is an important question. To examine this, we performed an additional analysis investigating whether there were any systematic differences in pulse frequency or pulse amplitude that may have led to the changes in neural drift rate. We did not find that either frequency or amplitude of stimulation resulted in a consistent effect on the neural drift rate. We also visually examined the anatomic distribution of stimulation sites, and did not observe any consistent patterns that could provide insight into why stimulation results in increases in drift rate on some occasions, and decreases on others. We now report these data in the revised Results and in Supplementary Figs. S6 and S7. The triggering protocol we used for stimulation was identical for all stimulation sessions.

Results

The effects of stimulation on drift rate could not be explained by stimulation parameters or site of stimulation (Supplementary Figs. S6 and S7).

3.5. What is the purpose of the closed-loop stimulation? Specifically of relevance here, did, at the points of time when the decoder decided whether to stimulate or not, the drift rates differ? Since the stimulator already predicted ‘bad performance’ when it was turned on, might this be what triggered the stimulation?

The motivation for conducting closed-loop stimulation was based on results from Ezzyat et al., 2017 (ref. 25) suggesting that positive effects of electrical stimulation on memory occur during trials in which the brain is in a poor encoding state and not during trials in which the brain is in a good encoding state. The closed-loop protocol used for these data was specifically designed to trigger stimulation when a classifier, based on distributed patterns of spectral power across multiple sites, determined that the brain was in a poor encoding state. We agree with the Reviewer that the question of whether the output of the classifier on different lists may have been linked to the drift rate is interesting. To address this, we conducted an analysis examining the correlation between drift rates and classifier output across lists. This analysis, now described in the Supplementary Results, did not reveal any consistent relationship. However, we note that this may be expected. The classifier was designed to capture trial-specific activity that is predictive of good encoding, yet neural drift describes a between-trial phenomenon. We therefore performed an additional analysis in which we explicitly looked at the individual instances of stimulation, and calculated a surrogate measure for drift rate by asking how similar the distributed pattern of neural activity was during that epoch compared to the activity in the previous epoch. We found that during instances of stimulation, there was no significant difference in similarity compared to trials in which stimulation was not applied. We have now described these analyses and results in Supplementary Fig. 10.

4. The manuscript is unnecessarily short, with many essential analysis and figures moved to supplement. It is unclear to me why that is. For example, S3 should clearly be part of Fig 3, otherwise the not sig different from zero effect for free recall is confusing. Same for S4, a needed analysis. All this needs to be integrated into one complete manuscript of the kind expected for this journal.

We thank the Reviewer for this suggestion, and we have moved several analyses and results, including those mentioned by the Reviewer, into the main text.

5. Statistics. There are several instances in the manuscript of “is of the opposite sign” when it is in fact not significantly different from zero. That is not a statistically valid conclusion. This concerns in particular the correlation of performance with drift in the free recall task, which on average is not significantly different from zero (except in S3 when using the patients that cluster well). This also applies to stimulation during free recall. There are of course statistical tests for sign changes, but the authors dont use these.

We agree with the Reviewer that the claim that the results of the free recall analysis are of the opposite sign is not valid. We have removed such claims from our manuscript, and instead focused our claims on the differences observed between the two tasks.

Reviewer #2 (Remarks to the Author):

The paper “Changing temporal context in human temporal lobe promotes memory of distinct episodes” by El-Kalliny et al. reports on intracranial EEG recordings from the temporal lobe in 78 epilepsy patients performing a memory task involving either free recall of a learned word list or, alternatively, cued recall from a learned list of word pairs. Focusing on the role of context during memory encoding, the authors quantify the amount change in power-spectral similarity from one trial lag to two lags as neural drift rate. Hypothesizing that changing context as measured by neural drift could be beneficial for separating distinct memories such as word pairs, but detrimental for associating sequential items in a word list, they indeed find the neural drift rate during encoding to be positively correlated with retrieval performance in the paired-associate task, but negatively correlated during the free recall task. In addition, they perform intracranial electrical stimulation in some patients. The resulting changes in neural drift across patients include both increase and decrease, but generally confirm the expected effects on the different memory tasks. The authors argue that their findings reconcile theories on temporal context as a binding feature and computational models on the role of temporal context in pattern separation.

Intracranial in-vivo recordings from the human brain are rare and precious, and the extraordinarily large number of subjects included in this study can only be obtained in a multi-center effort, which makes it even more special. The methods used in the study appear to be sound, although little bits of information are missing here and there. The paper is very well written and comes from a group well established in the field of human intracranial recordings. There is a lot to be liked about this paper. Nonetheless, I have a few concerns that should be addressed before publication.

We thank the Reviewer for these encouraging comments.

Major points.

(1) Some important information is missing in the paper. For instance, we are never explicitly told how many patients were eventually included in the analyses. Buried deep in the Supplement, we learn that subjects “in whom less than 10 electrode contacts were localized to the temporal lobe” were excluded from the analysis, but we are not told how many. Judging from the degrees of freedoms indicated in the t-test results ($t(74)$ and $t(27)$), 2 patients from the word-pair group seem to have been excluded, leaving 28 vs. 48 in both groups. However, statistical test descriptions in the SI Results ($t(48)$, $t(28)$, $t(76)$) are inconsistent with this. In the caption of Fig. 1, the authors mention “28 participants who performed the paired associates task, and 47 participants who performed the free recall task”, which again is inconsistent. The correct and final number of subjects included in the analysis should be made explicit right from the beginning. The authors may mention in the SI

that the original group size was even larger, but subjects had to be excluded for certain reasons. Reading about 78 subjects and finding out much later that only 76 or 75 of them were included in the relevant analyses is unnecessarily misleading. Also, we never learn how many sessions were actually recorded and analyzed. The total number, mean, and range across patients should be given, separately for sessions with and without stimulation, so we know exactly how many sessions were included in which analysis. When mentioning in the SI that “all experimental sessions in which a participant completed less than 20 lists” were excluded, the authors should state how many these were.

We apologize for any confusion regarding these points. We have now clarified these participant numbers in the revised Methods and throughout the text. Briefly, in total we initially started with 86 total participants. 33 performed the paired associates task, and 53 performed the free recall task. However, we excluded participants from our original dataset based on two criteria. First, we excluded from our analyses all experimental sessions in which participants completed less than 20 lists of the paired associates task. This resulted in the complete exclusion of two participants from the paired associates data. Second, we excluded from our analyses all participants in whom there was insufficient electrode coverage (< 10) in the temporal lobe to generate reliable estimates of distributed power across multiple electrode sites. This resulted in the exclusion of three participants from the paired associates data and five participants from the free recall data. How this second criteria affected our final data set, and the removal of participants from our initial dataset, was not made clear in our initial manuscript, and we apologize for that confusion.

In total, we retained 76 participants for our final analyses, 28 of whom performed the paired associates task and 48 of whom performed the free recall task. We have clarified this in the revised Methods, and have introduced the Results by starting with this final dataset. In addition, as part of this clarification, we have also clarified exactly how many experimental sessions were retained for analyses, which we now report in the Methods, Behavioral Tasks. Finally, we checked and revised all reported degrees of freedom so that they are consistent with this final dataset.

Methods – Participants

86 participants with drug resistant epilepsy underwent a surgical procedure in which platinum recording contacts were implanted subdurally on the cortical surface as well as deep within the brain parenchyma. We initially started with a dataset of 33 participants who performed a paired associates task and 53 participants performed a free recall task (Fig. 1A,B). However, we excluded data from two participants who performed the paired associates task because they did not complete a sufficient number of experimental lists to meet our inclusion criteria (see Behavioral Tasks). In addition, we excluded data from an additional three participants who performed the paired associates task and five participants who performed the free recall task because of insufficient electrode coverage in the temporal lobe of those participants (see Intracranial EEG Recordings). After excluding these participants from our initial dataset, we analyzed data from the 76 retained participants, 28 of whom performed the paired associates task and 48 of whom performed the free recall task.

Methods – Behavioral Tasks

For both tasks, we excluded all experimental sessions from our initial set of participants in which participants completed less than 20 lists (11 paired associates and 15 free recall sessions). This resulted in the complete exclusion of two participants who performed the paired associates task. The final dataset that we retained for our analyses of the experimental sessions that were performed without electrical stimulation included 28 participants who performed the paired associates task (63 experimental sessions, 2.33 ± 1.17 sessions per participant) and 48 participants who performed the free recall task (138 experimental sessions, 2.85 ± 1.61 sessions per participant). The final dataset that we retained for our analyses of the experimental sessions that were performed with stimulation included 7 participants who performed the paired associates task (11 unique stimulation sites, 1.18 ± 0.112 experimental sessions at each site) and 8 participants who performed the free recall task (12 unique stimulation sites, 1.75 ± 0.233 experimental sessions at each site).

Methods – Intracranial EEG (iEEG) Recordings

Following the exclusion of these electrodes, we then excluded all participants from our initial set of participants in whom less than ten electrode contacts were localized to the temporal lobe in order to ensure that the distributed pattern of oscillatory power across this brain region was well represented. This resulted in the exclusion of three participants from our initial dataset who performed the paired associates task, and five participants who performed the free recall task.

Results

We analyzed intracranial EEG (iEEG) data from 76 participants (41 male; age 36.2 ± 1.32 years; mean \pm SEM) with drug resistant epilepsy who underwent a surgical procedure for placement of intracranial electrodes for seizure monitoring, and then participated in either a verbal paired associates ($n=28$) or free recall task ($n=48$) task (Fig. 1A,B; see Materials and Methods).

On a similar note, instead of mentioning task descriptions like “up to 25 lists”, “up to 150 word pairs”, or “up to 300 total words”, it would be more informative to give the mean and range across sessions, again separately for sessions with and without stimulation. Finally, for the number of included electrode contacts, stating the range across patients would be informative.

We have now clarified the total number of experimental sessions, and the mean and standard deviation across participants, for all experimental tasks as described above. We have also clarified the number of electrodes contacts used in each participant for each task by reporting the mean and standard deviation of the number of contacts across participants.

SI Methods – Behavioral Tasks

Non-stimulation sessions consisted of 24.48 ± 1.42 lists, and stimulation sessions all consisted of 25 lists. (Paired associates)

Non-stimulation sessions consisted of 23.05 ± 2.56 lists, and stimulation sessions all consisted of 25 lists. (Free recall)

Methods – Intracranial EEG (iEEG) Recordings

We recorded from $128.2 \pm 6.57\%$ electrodes for each participant in paired associates, and from $141.6 \pm 5.60\%$ electrodes for each participant in free recall.

(2) The results from the electrical stimulation paradigm are quite intriguing. It would strengthen the authors conclusions if they could explicitly show that changes in neural drift rate caused by electrical stimulation explain changes in memory performance better than the presence of electrical stimulation per se.

We have now conducted three additional analyses to help clarify this point, and complement our initial analyses examining the effect of stimulation at the level of each participant. First, we examined whether changes in baseline similarity caused by electrical stimulation explain changes in memory performance. The failure to reject the null suggests that there is something unique about the effect of stimulation on neural drift that explains memory performance. Second, we examined the effects of stimulation on each list in each experimental session. We correlated the effect of stimulation on drift rate with the effect of stimulation on performance and across subjects, we found that there was a significant correlation. Third, we divided all lists into the tercile in which stimulation most increased drift rates and the tercile in which stimulation most decreased drift rates. We averaged the effect of stimulation on performance in these lists, and across participants, found that increases in drift rates were significantly associated with increases in performance. We now present these results in the revised text.

Results

We did not find a significant relation between stimulation-induced changes in baseline similarity of neural activity and stimulation-induced changes in memory performance, suggesting that it is specifically the rate of neural drift which supports successful formation of distinct memories (Supplementary Fig. 8).

...

Given the results observed during passive recordings of iEEG activity, we hypothesized that stimulation-induced increases in drift rate would be associated with stimulation-induced increases in performance, and vice versa. We correlated the effect of stimulation on performance with the effect of stimulation on drift rate across the 11 stimulation lists of each paired associates session (example; Fig. 4C). We found that there was a consistent correlation between effect of stimulation on performance and effect of stimulation on drift rate across participants

(Fig. 4D, top panel; $r = 0.175 \pm 0.070$, $t(10) = 2.52$, $p = 0.030$). For each stimulation session, we then divided the stimulation lists into terciles based on the effect of stimulation on drift rate. We found that memory performance was significantly improved in lists in the upper tercile compared to the lower tercile across participants (Fig. 4D, bottom panel; $12.27 \pm 4.35\%$, $t(10) = 2.82$, $p = 0.018$).

Minor points.

(1) In the Results section the authors state “The extent to which any pair of epochs had similar activity significantly decreased as more time elapsed between them” and then state results from an ANOVA to statistically undermine their statement. However, an ANOVA does not test for an increase or decrease or any directed relationship. Please use an appropriate statistical test (e.g., Spearman correlation).

We think that this is a good suggestion, and we have now used a Spearman correlation to examine the relation between neural similarity and elapsed lag between items. We report the mean correlation coefficient across participants, which demonstrates that on average, similarity decreases as the number of intervening lags increases.

Results

*The extent to which any pair of epochs had similar activity significantly decreased as more time elapsed between them (Spearman correlation between similarity and lags; mean correlation coefficient across participants for paired associates $r = -0.315 \pm 0.084$, $t(27) = -3.73$, $p = 5.3 * 10^{-4}$, one-sample; free recall $r = -0.436 \pm 0.047$, $t(47) = -9.20$, $p = 3.6 * 10^{-14}$).*

*The extent to which any pair of epochs had similar activity significantly decreased as more time elapsed between them (Spearman correlation between similarity and lags; paired associates, $r = -0.711 \pm 0.049$, $t(27) = -14.62$, $p = 2.39 * 10^{-14}$, one-sample; free recall, $r = -0.719 \pm 0.026$, $t(47) = -27.2$, $p = 2.01 * 10^{-30}$).*

(2) It is not entirely clear to me whether the analyses related to Figs. 1 and 2 included exclusively “record-only sessions”. Perhaps this could be made clearer.

We have now clarified that in our initial analysis, we first focused on experimental sessions in which we passively recorded iEEG activity as participants performed the two tasks. We then subsequently clarify that electrical stimulation provides an opportunity to examine the causal relation between neural drift and memory performance.

Results

We first focused on experimental sessions in which we passively recorded iEEG data while participants performed the tasks.

...

Our data examining passive recordings of iEEG activity reveal a relation between the rate of neural drift and memory performance in the paired associates task. Direct electrical stimulation of the brain, however, provides an opportunity to investigate the causal nature of this relation between neural drift and memory performance.

(3) For the data shown in Fig. 1E and related analyses, it is not clear to me over which time and frequency range the data was averaged. Please clarify.

We apologize that this point may have caused any confusion. Briefly, we initially hypothesized that the distributed power across the temporal lobe as captured by the iEEG data during the interstimulus epoch could provide the most direct representation of temporal context. However, we had no a priori assumptions about which frequencies were most relevant for this representation. As such, we constructed our initial feature vectors using data from the entire interstimulus epoch (averaged from -750 to 0 ms) and averaged over five frequency bands. We have now clarified this point in the revised Methods and Results

Methods

In our initial analysis of overall levels of neural drift, we used the power averaged over the interstimulus period (-750 to 0 ms relative to the presentation of each item) to calculate the average z-transformed power for each frequency, and then constructed a feature vector for each item by combining values of power across five frequency bands (Fig. 1): theta (3-8 Hz), alpha (8-12 Hz), beta (13-25 Hz), low gamma (30-58 Hz), and high gamma (62-100 Hz) (9).

Results

We hypothesized that activity during the epochs in which there is no visual stimulus may provide the most direct representation of a changing temporal context. As such, we constructed feature vectors of the distributed power across the temporal lobe using the instantaneous spectral power captured during the interstimulus epochs (-750 to 0 ms before item presentation; see Methods; Fig. 1C,D). Because we had no a priori assumptions about which frequencies may contribute to the representation of temporal context, we used the spectral power across five frequency bands to construct these feature vectors.

(4) p. 8/9: “we [...] constructed feature vectors using the z-transformed power averaged over the interstimulus epoch (-750 to 0 ms) and over 3 to 12 Hz” Was this entire range used or only time-frequency windows belonging to the significant cluster in Fig. 2B?

We also apologize that this point was not clearer in our initial manuscript. In order to identify which frequencies and time points were responsible for any relation between neural drift and

memory performance, we performed an analysis described in our manuscript where we constructed feature vectors separately using every individual frequency and every individual 500 ms window of time during the encoding period. We found a significant relation between neural drift and memory performance that was specific to the frequencies 3-12 Hz and to the interstimulus epoch. Based on this result, we then focused all post-hoc and subsequent analyses, including analyses comparing the passive record only dataset in paired associates to free recall and the analysis on the effects of electrical stimulation, on this interstimulus time period (-750 to 0 ms) and on this entire range of low frequencies (3-12 Hz). We have now clarified this in the revised Methods.

Methods

Then, in order to determine whether the relation between neural drift rate and memory performance was specific to individual frequencies and time points, we separately constructed feature vectors using each combination of individual frequencies and sliding 500 ms window (100 ms steps, 80% overlap; Fig. 2B). In post-hoc and stimulation analyses, we specifically focused on low frequency activity in the interstimulus epoch, and therefore constructed feature vectors using the z-transformed power averaged over the entire interstimulus epoch (-750 to 0 ms) and over all frequencies between 3 and 12 Hz.

Results

In post hoc analysis, we found that a significant correlation between neural drift computed using 3-12 Hz activity in the interstimulus period (-750 to 0 ms) across participants ($r = 0.095 \pm .033$, $t(27) = 2.86$, $p = .008$, one-sample; Fig. 2C).

(5) p.10: “we computed the rate of neural drift in the activity of interest.” What was this activity of interest? The same range as in point (4)?

The Reviewer is correct. We have edited the text to clarify this point:

Methods

In post-hoc and stimulation analyses, we specifically focused on low frequency activity in the interstimulus epoch, and therefore constructed feature vectors using the z-transformed power averaged over the entire interstimulus epoch (-750 to 0 ms) and over all frequencies between 3 and 12 Hz.

Results

For each participant receiving stimulation during the paired associates task (N=7, 11 unique stimulation sites), we z-transformed the rate of neural drift in interstimulus 3-12 Hz activity and memory performance on each stimulation list, relative to the distributions of drift rate and performance observed on non-stimulation lists.

(6) Related to the previous 3 points, the authors should go over their paper and make sure that for each and every neural drift rate calculated, it is absolutely clear from which time-frequency range it was derived.

We appreciate the Reviewer noting these points of confusion, and we have now revised our manuscript to make these points of analyses clear throughout.

(7) For the results shown in Figs. 3C, S3, S4, results of the one-sample t-test against 0 should also be given in cases where they were not significant in order to convey a feeling of whether there was a statistical trend or not.

We think this is a good suggestion and have now reported all of the results of the one-sample tests in the revised Results.

(8) p.9: “For participants who completed more than one session, we defined the participant level correlation as the average of the correlations across sessions.” This sentence comes before the first mention of the Fisher-z transform. Did the authors use the transform for this average also?

We applied the transform after calculating the participant-level averages. We have edited the revised Methods to clarify this point:

Methods

For participants who completed more than one session, we applied the z-transform after calculating the average correlation coefficient across sessions.

(9) p.10: “In each of 1000 permutations, we randomly inverted the sign of the correlation coefficient in each time-frequency point in each participant.” Was this a random permutation of existing signs or a random recreation of each single sign?

This was a random permutation of existing signs. We have edited the text to clarify this.

Methods – Statistical analysis

In each of 1000 permutations, we randomly flipped the sign of each correlation coefficient in each time-frequency point in each participant.

(10) In the SI, the authors describe the Morlet wavelets they used once as “wavelet number = 5” and once as “wave number = 5”. Is this number the “number of cycles”?

The Reviewer is correct, this is the number of cycles. We have reworded this for clarity.

Methods – Intracranial EEG (iEEG) Recordings

We convolved the continuous time iEEG recording from each electrode with 43 logarithmically spaced complex valued Morlet wavelets (wave number = 5 cycles) ranging from 3 to 100 Hz.

(11) SI Anatomical Localization: A common electrode target in the temporal lobe is the amygdala. Were there electrode contacts in the amygdala, and if so, were they excluded for a particular reason?

We thank the Reviewer for noting this error. There were indeed electrode contacts in the amygdala that were included in the analysis, but were mistakenly excluded from the text. We thank the reviewer for noting this error. We have clarified the localization of the electrodes in the revised Methods.

Methods – Anatomic Localization

For our analyses, we defined the lateral temporal lobe to consist of electrodes localized to inferior, middle, or superior temporal gyrus, and the medial temporal lobe to consist of electrodes localized to the parahippocampal gyrus, perirhinal cortex, entorhinal cortex, hippocampus, or amygdala.

(12) SI Closed Loop Electrical Stimulation: The authors state that to determine a safe stimulation amplitude, they went up to 1.5 mA for depth contacts. A few sentences later they state that in the paradigm they used up to 2 mA at depth contacts. Is this correct?

We have now clarified in the revised Supplementary Methods that the maximum stimulation amplitude was 2.5 mA for cortical surface contacts and 1.5 mA for depth contacts.

(13) SI Closed Loop Electrical Stimulation: “Data captured during non-stimulation lists were also used for the z-transform of input features.” Why “also”?

We used the first three lists during the stimulation sessions for normalization of the classifier inputs. In addition, we also used data captured during all subsequent non-stimulation lists for normalization of the classifier inputs. As the closed-loop stimulation sessions proceed in real-time, the classifier inputs are continuously updated and normalized. We had intended to make this point when we initially wrote that these data were also used for the z-transform of the input features. We have now clarified this point.

SI Closed Loop Electrical Stimulation

Stimulation was not applied during the first three lists, and baseline spectral data collected during these lists was used for the z-transform normalization of the input features. During non-stimulation lists, spectral power features and classifier output were computed identically to stimulation lists, but stimulation was disabled. Data captured during non-stimulation lists were used for continued z-transform normalization of input features.

(14) SI Closed Loop Electrical Stimulation: “In cases in which no lateral temporal cortex contacts were available, we selected an electrode pair at or near the largest SME elsewhere in the brain.” How many cases?

We have added this information to the Supplementary Methods.

SI Closed Loop Electrical Stimulation

In cases in which no lateral temporal cortex contacts were available, we selected an electrode pair at or near the largest SME elsewhere in the brain (two participants in paired associates, three participants in free recall).

(15) SI Closed Loop Electrical Stimulation: “For a subset of sessions, the particular amplitude and frequency used for stimulation were chosen based on a pre-test in which we stimulated the brain at each parameter combination, while the patient was at rest (no experimental task).” How many sessions?

We have added this information to the Supplementary Methods.

SI Closed Loop Electrical Stimulation

For a subset of participants, the particular amplitude and frequency used for stimulation were chosen based on a pre-test in which we stimulated the brain at each parameter combination, while the patient was at rest (no experimental task; 3 of 7 participants in paired associates, 7 of 8 participants in free recall). The frequency-amplitude combination that maximized the change in classifier output was used in the closed-loop memory task.

(16) SI Results: The text referring to Fig. S1 is a bit confusing. Maybe switching panels A and B would help.

We have made this change and changed the text accordingly.

Typos.

-SI, p.2: “Based on this criteria” should be “Based on this criterion” or “Based on these criteria”.

-SI, p.3: “we trained a logistic regression classifier trained to ...” Omit the second “trained”.

Thank you for pointing out these errors. We have made these corrections.

Reviewer #3 (Remarks to the Author):

This manuscript addresses the question whether changes in temporal context in terms of spectral EEG patterns within the temporal lobe influence verbal memory performance. The analyses are based on intracranial EEG recordings in 78 presurgical epilepsy patients,

performing either a paired associates task or a free recall task. The authors hypothesized that decreased spectral similarity of inter-stimulus EEG during encoding is related to increased performance in the paired associates task, because spectral similarity may impede pattern separation. On the other hand, they supposed that free recall performance may rather be facilitated by context similarity. As major results the authors report that the rate of drift of low frequency EEG patterns within temporal lobe indeed is positively correlated with performance in the paired associates task. This relation is found to be opposite for the free recall task. Finally, the authors show that context similarity can be modulated by electric stimulation and that stimulation-related changes in context similarity correspond to changes in memory performance.

In my view, the questions and hypotheses underlying this study are very timely and important for the understanding of human memory dynamics. The manuscript is based on a precious data set, i.e. a large number of intracranial EEG recordings in epilepsy patients performing verbal memory tasks. The analysis approaches and techniques appear highly sophisticated and up-to-date. Importantly, the electric stimulation sessions provide additional information regarding the causal relevance of the observed interrelations between context similarity and memory performance. The interpretation and discussion of the results appears straightforward and convincing. Thus, I have only a few comments, which are listed below:

Major:

- The authors present behavioral evidence for temporal contiguity effects in the free recall data (supplement). It should be considered whether also a measure reflecting the degree of cognitive separation in the paired associates task could be calculated, for instance, by evaluating intrusions (i.e. wrong associations belonging to other word cues). Similar to the results for the free recall task, this measure then could be related to the observed correlations between context similarity in terms of iEEG patterns and memory performance.*

We thank the Reviewer for suggesting this analysis. As the Reviewer notes, a direct measure of cognitive separation in the paired associates task is the extent to which a participant makes an intrusion. As such, we examined the relation between the degree of baseline similarity in each list and the number of behavioral intrusions and found a significant relation. We now include this result in the revised Results.

Results

The overlap in temporal context observed when memory performance is worse suggests that such overlaps could lead to errors during retrieval. In the paired associates task, a direct measure of such errors is the extent to which participants retrieve the incorrect word, thereby making an intrusion. We therefore hypothesized that greater overlap in temporal context throughout the encoding period as reflected by a greater baseline similarity would be related to greater rates of intrusions. Indeed, we found that across participants, the number of intrusions vocalized during lists with greater baseline similarity was significantly greater than the number of intrusions vocalized during lists with lower baseline similarity (1.59 intrusions +/- 0.236; $t(21) = 2.50$, $p = .021$; $n = 22$ participants who had vocalized intrusions on at least three lists).

• The authors implemented artifact rejection only in terms of a channel exclusion procedure. There still may be a considerable amount of trials contaminated by epileptiform activity in the remaining channels. Since proper artifact rejection is essential, in particular, when dealing with iEEG data from epilepsy patients, the authors should provide evidence that their results are not affected by epileptiform activity.

The Reviewer raises a valid concern regarding any analyses of intracranial electrode data. To address this, we performed additional procedures for excluding these possible electrodes and epileptiform artifacts. First, we now exclude from our analyses all electrodes determined to be in the seizure onset zones by clinicians at the various hospitals. Next, as we had reported previously, we remove all electrodes from our analyses that exhibit a kurtosis or variance that is two standard deviations from the mean. We now report how many electrodes were excluded based on this criterion, and how such exclusions affected the final number of participants retained for analyses. To supplement this, we now include an additional step for addressing this concern. For all retained electrodes, we performed an exclusion procedure conducted at the level of individual trials in which we removed all trials that exhibit kurtosis or variance that is greater than two standard deviations from the mean. This ensures that any epileptiform discharges or signal artifacts that are present on the retained electrodes do not adversely affect our results. We have re-analyzed our data and presented all of our results following these steps, and we have described these steps in the revised Methods.

Methods – Intracranial EEG (iEEG) Recordings

We minimized any confounding effects that may be related to transient epileptic activity (interictal discharges) by first removing all electrodes identified by clinicians as part of the seizure onset zone from further analysis. We then used an iterative cleaning procedure, in which we concatenated the raw voltage trace from all encoding epochs (-3000 ms to 6000 ms relative to the word pair presentation) separately for each electrode. We then eliminated electrodes with kurtosis or variance greater than two standard deviations from the persistent sample mean. Based on these criteria, we excluded 27.4 +/- 4.13 % of electrodes in each participant. Following the exclusion of these electrodes, we then excluded all participants from our initial set of participants in whom less than ten electrode contacts were localized to the temporal lobe in order to ensure that the distributed pattern of oscillatory power across this brain region was well represented. This resulted in the exclusion of three participants from our initial dataset who performed the paired associates task, and five participants who performed the free recall task. Finally, for each experimental session we then calculated the variance of the raw voltage trace for each encoding epoch, averaged across electrodes, and eliminated all trials with kurtosis or variance greater than two standard deviations from the persistent sample mean. Based on these criteria, we excluded 11.1 +/- 1.1 % of trials in each free recall session and 13.4 +/- 3.57 % of trials in each paired associates session.

• Do the authors observe any laterality effects, i.e. do the results differ when being separately evaluated for the left and the right hemisphere?

We examined the relation between drift rate and paired associates memory performance independently for temporal lobe electrodes in the left and right hemisphere, and found a significant and similar pattern of results for both. Left hemisphere: $t(23) = 2.81$, $p=0.010$. Right hemisphere: $t(26) = 2.61$, $p=0.015$.

Minor:

• **Page 4: “We found that the rate of drift ... was correlated ... across all participants”: please state the average correlation coefficient across subjects.**

We now report the average correlation coefficient across participants for all tests used in our analyses.

• **Page 5: “... the decision to stimulate was controlled by a classifier ...”. Which classifier features predicted successful memory retrieval?**

There is a consistent pattern of increases in high-frequency activity and decreases in low-frequency activity during successful memory encoding. We have previously reported these effects in both paired associates (Greenberg et al., 2015) and free recall (Ezzyat et al., 2017) datasets, and have reproduced those effects here using this dataset (Supplementary Fig. 3).

Supplementary Methods

In both tasks, successful memory formation is consistently accompanied by a pattern of increases in high-frequency activity and decreases in low-frequency activity during the encoding period (Supplementary Fig. 3) (Greenberg et al., 2015; Ezzyat et al., 2017). Classification of good memory encoding states in general identify when this pattern emerges in the neural data (Ezzyat et al., 2017).

• **Figure 1c: The timing of stimulus presentations during the encoding phase is quite different for the two paradigms. Please discuss whether or not this may have had an impact on the results.**

The Reviewer raises a good point. We have now conducted an additional analysis to examine whether the observed results may be confounded by the different lengths of the encoding epochs in the two tasks. Briefly, the encoding time for one word pair in the paired associates tasks is approximately equal to the time of two successive item presentations in free recall. We therefore examined the rate of neural drift in the free recall task using interstimulus epochs separated by two lags in that task, which approximately matches the interstimulus period between word pairs in the paired associates task. We found that the difference in the relation between drift rate and memory performance between the two tasks was relatively unchanged. We now report this additional analysis in the revised Results and in a new supplementary figure (Fig. S5).

Results

We also confirmed that the difference between paired associates and free recall was not related to the differing length of encoding epochs in the two tasks. We repeated our analysis by defining

a single unit of lag in free recall as interstimulus epochs that were separated by two intervening item presentations to match the elapsed time between word pairs in the paired associates task, and found a similar difference in the relation between drift rate and memory performance (paired associates vs free recall, $t(74) = 2.76$, $p = .0072$, two-sample; free recall, $t(47) = -1.43$, $p = .159$; Supplementary Fig. 5).

• Supplement, page 5: “We found no significant clusters ...”. In my view, this result regarding the relation between iEEG pattern drift and free recall performance should be reported in the main text.

We agree with the Reviewer and have now reported this result in the main text.

Results

We confirmed that there was no other time-frequency cluster of activity at which the rate of neural drift was correlated with free recall memory performance (Supplementary Fig. 4).

• Supplement, page 5: “The tertile of participants ... demonstrated a relation ...”. Again, in my view, this result should be reported in the main text.
• Supplement, page 5: “We divided the participants ... averaged across all recalls (0.664 ± 0.0078). Please add statistics testing whether this temporal contiguity value is above chance (i.e. above 0.5).

We also agree with the Reviewer that this is an important result to report, and have now moved this analysis to the main text of our revised Results. We have reported that the average temporal contiguity during free recall was greater than chance. We have also included the results of this analysis in our revised Figure 3.

Results

*We hypothesized that the observed differences between paired associates and free recall could be driven by the subset of free recall participants who demonstrate a behavioral tendency to bind items across time in the service of improved memory performance. To investigate this possibility, for each participant we derived a value of temporal factor, which reflects the degree to which freely recalled items tend to be linked to one another in time, or clustered (see Methods) (21). Participants during free recall exhibited significant temporal clustering averaged across all recalls (0.665 ± 0.013 , $t(47) = 12.5$, $p = 1.37 * 10^{-16}$). We divided the participants into an upper and lower tertile based on this value. The tertile of participants demonstrating the highest temporal clustering during recall (0.759 ± 0.019 , $n=16$) demonstrated a relation between neural drift rate and performance that was significantly different from the relation observed during paired associates (free recall, $t(15) = -1.91$, $p = .076$; paired associates vs free recall $t(42) = 3.31$, $p = .002$; Fig. 3). Conversely, the tertile of participants demonstrating the lowest temporal clustering (0.575 ± 0.011 , $n=16$) did not (free recall, $t(15) = 1.00$, $p = .33$; paired associates vs free recall, $t(42) = 0.623$, $p = .54$).*

Thank you again for taking the time to consider our manuscript for publication. We have included a revised version of the manuscript with this submission. We look forward to your reply and to the reviews of our manuscript.

Sincerely,

Kareem A. Zaghoul, MD, PhD

REVIEWERS' COMMENTS:

Reviewer #1 (Remarks to the Author):

The authors prepared a careful revision and addressed all my concerns. I support publication in this form.

Reviewer #2 (Remarks to the Author):

The authors have addressed all my points and have made the necessary clarifications in the text. I look forward to seeing this nice paper published.

Florian Mormann

Reviewer #3 (Remarks to the Author):

The authors have satisfactorily addressed all of my comments.

I have just one remaining minor remark:

Page 5: Indeed, we found that across participants ... (1.59 +/- 0.236 intrusions; $t(21) = 2.50$... Both intrusion averages should be reported here.